# Engineering multifunctional rhizosphere probiotics using consortia of *Bacillus amyloliquefaciens* transposon insertion mutants

Jingxuan Li[1†], Chunlan Yang[1†], Alexandre Jousset[1], Keming Yang[1], Xiaofang Wang[1], Zhihui Xu[1], Tianjie Yang[1], Xinlan Mei[1], Zengtao Zhong[2], Yangchun Xu[1], Qirong Shen[1], Ville-Petri Friman[1,3]*, Zhong Wei[1]*

[1]Key Lab of Organic-based Fertilizers of China and Jiangsu Provincial Key Lab for Solid Organic Waste Utilization, Nanjing Agricultural University, Nanjing, China; [2]College of Life Science, Nanjing Agricultural University, Nanjing, China; [3]Department of Microbiology, University of Helsinki, Helsinki, Finland

**Abstract** While bacterial diversity is beneficial for the functioning of rhizosphere microbiomes, multi-species bioinoculants often fail to promote plant growth. One potential reason for this is that competition between different species of inoculated consortia members creates conflicts for their survival and functioning. To circumvent this, we used transposon insertion mutagenesis to increase the functional diversity within *Bacillus amyloliquefaciens* bacterial species and tested if we could improve plant growth promotion by assembling consortia of highly clonal but phenotypically dissimilar mutants. While most insertion mutations were harmful, some significantly improved *B. amyloliquefaciens* plant growth promotion traits relative to the wild-type strain. Eight phenotypically distinct mutants were selected to test if their functioning could be improved by applying them as multifunctional consortia. We found that *B. amyloliquefaciens* consortium richness correlated positively with plant root colonization and protection from *Ralstonia solanacearum* phytopathogenic bacterium. Crucially, 8-mutant consortium consisting of phenotypically dissimilar mutants performed better than randomly assembled 8-mutant consortia, suggesting that improvements were likely driven by consortia multifunctionality instead of consortia richness. Together, our results suggest that increasing intra-species phenotypic diversity could be an effective way to improve probiotic consortium functioning and plant growth promotion in agricultural systems.

*For correspondence:
ville-petri.friman@helsinki.fi (V-PF);
weizhong@njau.edu.cn (ZW)

†These authors contributed equally to this work

Competing interest: The authors declare that no competing interests exist.

## Editor's evaluation

The work is significant because it reports that intra-species phenotypic diversity in bacteria could be an effective way to improve probiotic consortia in agriculture. Two solid conclusions emerge from this important work: (1) Communities of near-clonal bacteria can be optimized based on functional traits to promote functional diversity. (2) Consortium multifunctionality – but not richness – is key to promoting bacterial colonization in roots and to protection against a plant pathogen. The work has broad practical implications for designing probiotic consortia that promote host health beyond the plant-microbe interaction field.

## Introduction

Bacterial rhizosphere microbiome diversity has been strongly associated with beneficial effects on plant growth in both natural and agricultural environments (*Mendes et al., 2013*; *Yu et al., 2019*). Such beneficial diversity effects are often associated with increased functional diversity and community multifunctionality (*Saleem et al., 2019*; *Raza et al., 2020*; *Hu et al., 2021*; *Raza et al., 2021*; *Lindsay et al., 2021*), where different taxa play complementary roles in plant growth promotion, specializing in nutrient solubilization, plant immune priming or for example by interacting with beneficial or pathogenic rhizosphere microbes (*Hayat et al., 2010*; *Lugtenberg and Kamilova, 2009*). Moreover, high microbial diversity can have other benefits such as providing functional redundancy in the case of species extinctions (*Allison and Martiny, 2008*), community stability (*Hu et al., 2016*), or provide emergent consortia-level properties that cannot be predicted based on the sum of individual species (*Netzker et al., 2018*; *Abrudan et al., 2015*; *Tilman, 1982*; *Tilman, 2004*). While several studies have tried to harness positive effects of bacterial diversity for crop production (*Weidner et al., 2015*; *Hu et al., 2017*; *Wu et al., 2018*), diverse inoculants often fail to produce the desired benefits in field conditions. This discrepancy could rise due to constraints set by external factors, such as resource availability (*De-la-Peña et al., 2010*; *Gransee and Wittenmayer, 2000*; *Chaparro et al., 2013*), that constrains the expression of plant growth promotion traits, or the presence of competitors, parasites, and predators that reduce the survival of the inoculated bacteria in the rhizosphere (*Hibbing et al., 2010*; *Gao et al., 2019*). Also intrinsic factors, such as interactions between inoculant consortia members, could constrain their survival in the rhizosphere (*Wang et al., 2021*; *Jousset et al., 2011*; *Gu et al., 2020*). It has thus been suggested that microbial inoculant design should aim to minimize negative interactions within the inoculated consortia without compromising consortia competitive ability, which is important for consortia establishment in the diverse rhizosphere microbiome, especially when being initially rare (*Gu et al., 2020*; *Li et al., 2019*; *Wei et al., 2015*).

Diversity-ecosystem functioning experiments are widely used to study if communities perform better due to inherent benefits of diversity or because certain communities contain specific species that are important for the community functioning (i.e., species identity effects due to presence of certain microbial keystone species) (*Banerjee et al., 2018*; *Jousset et al., 2014*). In microbial context, such experiments have been mostly conducted by using taxonomically diverse communities, where associations between community diversity and functioning have been found to range from positive to neutral and negative (*Jousset et al., 2011*; *Wei et al., 2015*; *Maynard et al., 2017*). One reason for such variety of outcomes might be that increasing community diversity often also introduces competition between the community members, which might overshadow and constrain the benefits of inoculant diversity. For example, antimicrobial activity of pathogen-suppressing volatile organic compounds has been shown to peak at intermediate levels of bacterial community diversity (*Raza et al., 2020*), and a similar hump-shaped pattern has also been found between toxin production and bacterial community richness (*Jousset et al., 2011*). Likely explanation for these findings is that the same compounds that are responsible for pathogens suppression also affect the competition between other consortia members (*Gu et al., 2020*). One way to reduce this negative effect would be to quantify species interactions in advance to assemble multifunctional bioinoculants with weak within-consortia competitive interactions (*Li et al., 2019*; *Wei et al., 2015*). Such optimization could be achieved by using highly related bioinoculant strains because genetic relatedness is predicted to favor cooperation instead of conflict due to kin selection (*Simonet and McNally, 2021*; *Crespi, 2001*; *Kolter and Greenberg, 2006*; *Raymond et al., 2012*). For example, if bioinoculant species produce antimicrobials, it is likely that related strains share the same antibiotic biosynthesis and resistance genes, and hence, unlikely antagonize each other (*Xia et al., 2022*). Increasing intra-species diversity could further lead to increased bioinoculant consortia multifunctionality, which has been shown to play an important role in the functioning of microbial ecosystems (*Van Rossum et al., 2020*; *Fields et al., 2021*; *Raffard et al., 2019*; *Nicastro et al., 2020*; *Blake et al., 2021*; *Dragoš et al., 2018*). Multifunctionality could be optimized based on ecological complementarity by assembling consortia from strains that use different niches or specialize in different functions within the same niche (*Dragoš et al., 2018*; *Martin et al., 2016*). This might allow more efficient expression or production of different compounds at the consortia level and help to overcome any antagonistic pleiotropic effects experienced at the individual strain level (i.e., trade-offs between plant growth promotion traits). Moreover, interactions between near clonal specialist genotypes could allow division of labor, where shared

workload by two specialists leads to higher productivity relative to one generalist as has been demonstrated with *Bacillus subtilis* bacterium during biofilm matrix production (*Dragoš et al., 2018*). Such overperformance (or overyielding) could also be driven by other mechanisms such as resource partitioning, abiotic facilitation, or biotic feedbacks (*Barry et al., 2019*), leading to higher consortia functioning than predicted based on the sum of its individual members.

Here, we used a combination of genetics, molecular biology, and biodiversity-ecosystem functioning theory (*Saleem et al., 2019*; *Hu et al., 2016*) to test if increasing phenotypic diversity of a single plant growth-promoting *Bacillus amyloliquefaciens* T-5 bacterium offers a viable strategy to improve bioinoculant consortia multifunctionality. We chose *B. amyloliquefaciens* T-5 strain as our model species because it originates from the tomato rhizosphere (*Tan et al., 2013*) and has previously been shown to protect plants from various diseases, including bacterial wilt caused by phytopathogenic *Ralstonia solanacearum* bacterium (*Jiang et al., 2017*). To increase functional diversity within a single species, we first created a *B. amyloliquefaciens* mutant library using TnYLB-1 transposon mutagenesis (*Le Breton et al., 2006*), resulting in 1999 unique insertion mutants. A representative subset of 479 mutants were chosen for high-throughput phenotyping in vitro in the lab regarding four important plant growth promotion traits: biomass production, biofilm formation, swarming motility, and pathogen suppression (*Lugtenberg and Kamilova, 2009*; *Turnbull et al., 2001*; *Raaijmakers et al., 2002*). After phenotyping, a subset of 47 mutants that performed better or equally well as the wild-type strain were taken forward to plant experiments to determine if in vitro phenotyping could predict mutant success in terms of tomato root colonization and plant protection from *R. solanacearum* in vivo. Finally, we tested if plant growth promotion could be improved by combining phenotypically distinct mutants into multi-strain consortia. We predicted that increasing mutant consortia richness would lead to improved root colonization and plant protection if mutants are phenotypically dissimilar, which could result in positive diversity-ecosystem relationships or trait multifunctionality. These two hypotheses were tested by comparing the performance between phenotypically dissimilar and randomly assembled mutant consortia. Our results demonstrate that transposon insertion mutagenesis is an effective way to improve and identify genes underlying plant growth promotion traits in *B. amyloliquefaciens*. Importantly, we show that in vitro phenotyping can be used to optimize inoculant consortia functioning in vivo and that diverse mutant consortia are better at colonizing and protecting tomato plants when they are assembled based on phenotypic dissimilarity.

## Results

### Effects of transposon insertions on *B. amyloliquefaciens* T-5 mutant traits measured in vitro and in vivo

We first quantified the phenotypic effects of transposon insertions on *B. amyloliquefaciens* T-5 mutant traits across the whole mutant library (1999 mutants in total, *Figure 1—figure supplement 1*, *Supplementary file 1a*). To make the number of mutants more manageable for in vivo experiment, we randomly selected a subset of 479 mutants for further analyses (*Supplementary file 1b*). While we likely lost certain unique mutants in the process, the sampled subset was phenotypically representative of the original collection based on four measured traits (Mantel test; r=0.7591, p=0.04167). Within this subset, most insertions had negative effects on the four measured phenotypic traits, with more than half of the mutants showing reduced swarming (58.7%), biomass production (67.2%), and biofilm formation (60.8%) compared to the wild-type strain (*Figure 1A*). In contrast, the median effect of insertions affecting the pathogen suppression was neutral, and 51.1% of the mutants showed only a moderate increase in their suppressiveness (*Figure 1A*, *Supplementary file 1b*). In line with this finding, the distribution of effects of insertions on each trait was skewed, where beneficial mutations resulted mainly in a moderate improvement, while harmful mutations often led to severe reductions in measured traits (*Figure 1A*). Moreover, several insertions caused trade-offs, where improvement in one trait led to a reduction in another trait (*Figure 1B*). For example, swarming motility correlated negatively with biomass production, while biomass production led to a trade-off with both biofilm production and pathogen suppression (*Figure 1B*). These results thus suggest that transposon insertions constrained the simultaneous improvement of multiple traits, leading to specialized *B. amyloliquefaciens* T-5 mutants, which could be clustered in three phenotypic groups based on K-means clustering (Adonis test: $R^2$=0.5283, p<0.001, *Figure 1C*, *Figure 1—figure supplement 2*). Compared

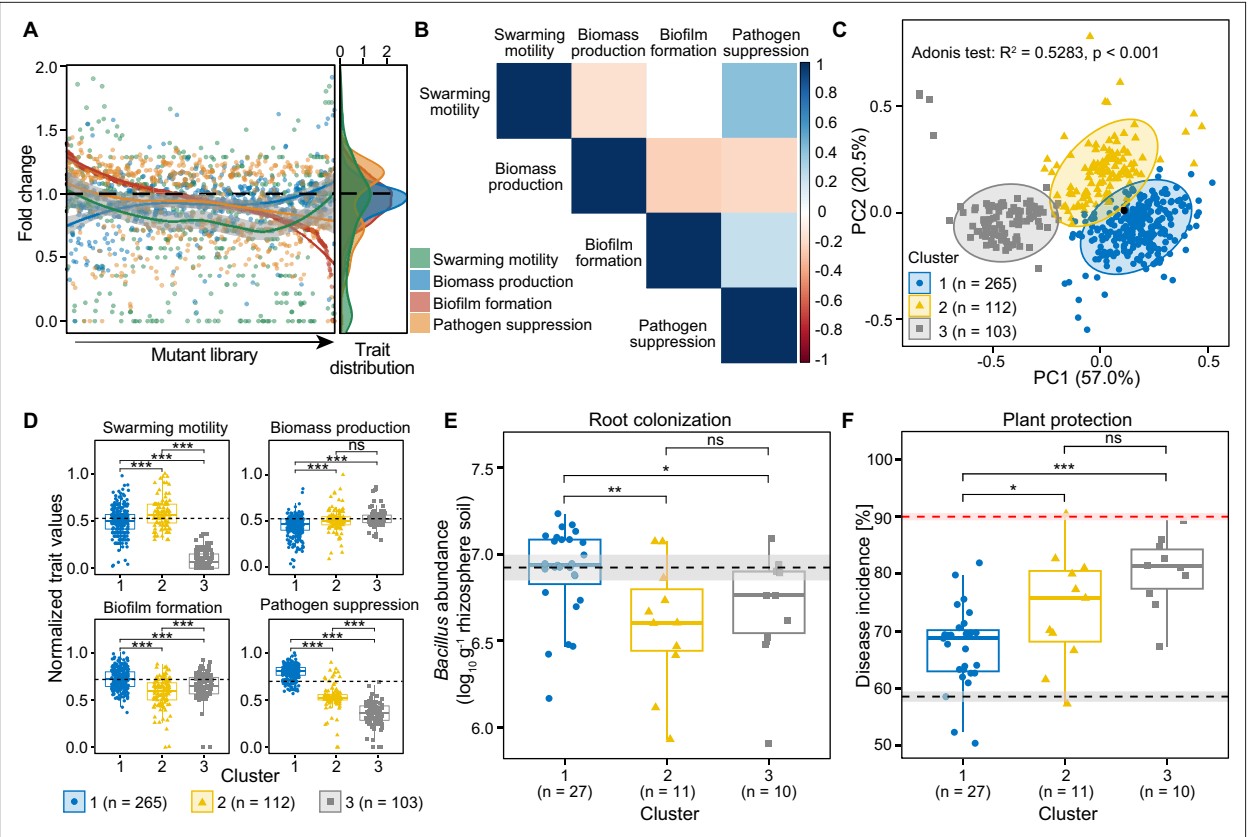

**Figure 1.** The effects of transposon insertions on the traits of 479 *B. amyloliquefaciens* T-5 mutants measured in vitro and in vivo. (**A**) shows distribution of fold changes regarding mutants' swarming motility, biomass production, biofilm formation, and pathogen suppression relative to the wild-type strain (black dashed line equaling 1). (**B**) displays pairwise covariance matrix between individual traits, where red cells indicate negative trait correlations (trade-offs) and blue cells positive traits correlations; white cells indicate no correlation between the given traits. (**C**) shows principal coordinates analysis showing the clustering of all mutants and the wild-type (black point). Mutants were assigned to different clusters based on K-means algorithm of four measured traits. (**D**) shows mean trait differences between clusters based on unpaired two-samples Wilcoxon test. The wild-type was assigned in the cluster 1 based on K-means clustering and its trait values are shown as dashed black lines. (**E**) displays root colonization of representative 47 *B. amyloliquefaciens* mutants from clusters 1–3 relative to the wild-type strain (black dashed line) based on cell densities in the root system 30 days post pathogen inoculation (dpi). (F) shows plant protection of representative 47 *B. amyloliquefaciens* mutants from clusters 1–3 relative to the wild-type strain (black dashed line), and negative 'pathogen-only' control (red dashed line), quantified as bacterial wilt disease incidence 30 dpi. Shaded areas in (**E** and **F**) represent the mean ± SEM. Since (**D**) displays the normalized trait values, variation for the wild-type strain is not shown. Pairwise differences in (**D–F**) were analyzed using unpaired two-samples Wilcoxon test: *** denotes for statistical significance at $p<0.001$; ** denotes for statistical significance at $p<0.01$; * denotes for statistical significance at $p<0.05$; ns denote for no significance.

The online version of this article includes the following figure supplement(s) for figure 1:

**Figure supplement 1.** The effects of transposon insertions on the traits of 1999 mutants included in the original library measured in vitro.

**Figure supplement 2.** Optimal number of K-means clusters suggested by gap statistic method.

**Figure supplement 3.** Identification of disrupted genes of 47 transposon insertion mutants based on B. amyloliquefaciens T-5 reference genome.

**Figure supplement 4.** Positive and linear relationships between optical density ($OD_{600\,nm}$) and cell counts (colony forming units [CFUs]) of wild-type (black), M78 (representative mutant with decreased biomass production, red) and M109 (representative mutant with increased biomass production, blue) strains.

to the other two clusters, mutants belonging to the cluster 1 showed significant increases in biofilm formation and pathogen suppression but reduced biomass production (***Figure 1D***, ***Supplementary file 2a***). Mutants in the cluster 2 showed improved swarming motility and reduced pathogen suppression, while mutants in the cluster 3 had poor performance overall, showing highly reduced swarming motility and pathogen suppression (***Figure 1D***, ***Supplementary file 2a***).

To test if the mutants clustered in different phenotypic groups also differed in their tomato root colonization and ability to protect plants from *R. solanacearum* infections, 47 mutants representing

a subset of three clusters were randomly selected for a greenhouse experiment (the specific effects of insertions on biological processes, cellular components, and molecular function for all mutants are shown in *Figure 1—figure supplement 3B* and *Supplementary file 1c*). Compared to the wild-type, 57% of these mutants (27/47) reached lower population densities in the rhizosphere (30 days post-pathogen inoculation [dpi]), and this was especially clear for mutants belonging to clusters 2 and 3. In contrast, mutants belonging to the cluster 1 retained more efficient root colonization compared to mutants from the other clusters, and more than half (16 of 27) of the cluster 1 mutants showed improved root colonization relative to the wild-type (30 dpi, *Figure 1E*, *Supplementary file 2b*). Around 93% of all mutants (44/47) exhibited reduced plant protection relative to the wild-type strain. However, mutants from the cluster 1 showed higher plant protection compared to the other two clusters, and specifically, two of the cluster 1 mutants showed improved plant protection relative to the wild-type strain (30 dpi, *Figure 1F*, *Supplementary file 2b*). Together, these results show that while most transposon mutants had reduced performance relative to the wild-type strain, some of the

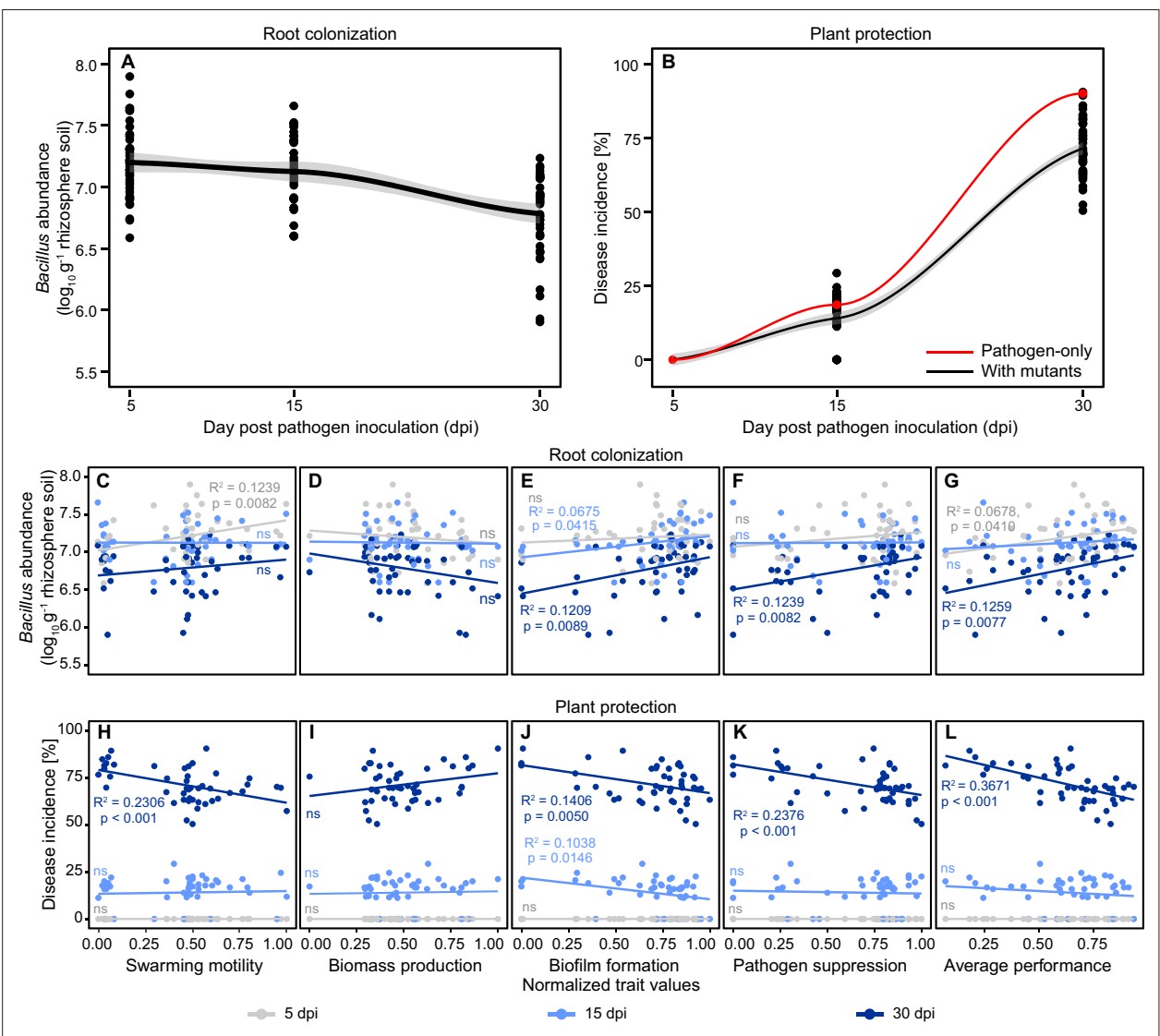

**Figure 2.** Regression analysis explaining root colonization and plant protection of representative 47 *B. amyloliquefaciens* mutants based on their trait values measured in vitro at different sampling time points. (**A** and **B**) show the dynamics of root colonization and plant protection, respectively. The red and black lines in (**B**) show the disease incidence of pathogen-only control and *B. amyloliquefaciens* mutant treatments, respectively. (**C–G**) and (**H–L**) show root colonization and plant protection, respectively, correlated with different traits at 5 days post pathogen inoculation (dpi) (gray), 15 dpi (light blue), and 30 dpi (dark blue) time points. Significant relationships and R-squared values are shown in panels with colors corresponding to the sampling time points ('ns' denotes for non-significant relationship).

**Table 1.** Analysis of variance (ANOVA) table summarizing the effects of mutant traits measured in vitro on the root colonization and plant protection.

Separate models were run for each dependent variable at different time points (5, 15, and 30 days post pathogen inoculation [dpi]) and all response variables were treated as continuous variables (bacterial abundances were log-transformed before the analysis). Table data represent only the most parsimonious models based on the Akaike's information criterion (AIC) where 'NA' denotes variables that were not retained in the final models, 'df' denotes degrees of freedom, and '$R^2$' denotes total variance explained by regression coefficient of determination. The arrows represent the direction of coefficient values: ↑: coefficient >0; ↓: coefficient <0. Significant effects (p<0.05) are highlighted in bold.

| | | | | | | | | | |
|---|---|---|---|---|---|---|---|---|---|
| | | | | Day post pathogen inoculation (dpi) | | | | | |
| | | 5 dpi | | | 15 dpi | | | 30 dpi | |
| Mutant trait | df | F | p | df | F | p | df | F | p |
| Root colonization (*Bacillus* abundance – log CFU g$^{-1}$ rhizosphere soil) | | | | | | | | | |
| Swarming motility | 1 | **5.32** | **0.0259**↑ | 1 | 0.006 | 0.9404↑ | 1 | 0.008 | 0.9302↑ |
| Biomass production | 1 | 0.46 | 0.5032↓ | 1 | 0.58 | 0.4512↑ | 1 | 0.07 | 0.7886↓ |
| Biofilm formation | 1 | 0.69 | 0.4103↑ | 1 | **4.34** | **0.0433**↑ | 1 | **5.64** | **0.0220**↑ |
| Pathogen suppression | 1 | 2.04 | 0.1609↓ | 1 | 0.006 | 0.9410↓ | 1 | **8.10** | **0.0068**↑ |
| | | $R^2$=0.0875 | | | $R^2$=0.0193 | | | $R^2$=0.1730 | |
| No. of residuals | 43 | AIC: 16.18 | | 43 | AIC: 3.82 | | 43 | AIC: 25.06 | |
| Plant protection (disease incidence [%]) | | | | | | | | | |
| Swarming motility | NA | NA | | 1 | 0.34 | 0.5620↑ | 1 | **6.89** | **0.0119**↓ |
| Biomass production | NA | NA | | 1 | 0.94 | 0.3369↓ | 1 | 0.23 | 0.6339↑ |
| Biofilm formation | NA | NA | | 1 | **5.71** | **0.0213**↓ | 1 | **12.34** | **0.0016**↓ |
| Pathogen suppression | NA | NA | | 1 | 0.14 | 0.7090↓ | 1 | **20.90** | **<0.001**↓ |
| | | NA | | | $R^2$=0.0625 | | | $R^2$=0.4294 | |
| No. of residuals | | NA | | 43 | AIC: –101.24 | | 43 | AIC: –112.38 | |

mutants showed improvement in at least in one plant growth promotion trait, which often resulted in trade-offs with some other traits.

## Phenotypic trait variation explains mutant success in rhizosphere colonization and plant protection in vivo with tomato

To test if phenotypic trait variation measured in vitro correlates with beneficial effects on plants in vivo, the rhizosphere colonization and plant protection of 47 phenotypically distinct mutants was quantified after 5, 15, and 30 dpi in a greenhouse experiment. Results showed that *B. amyloliquefaciens* inoculations led to approximately 20.6% mean reduction in bacterial wilt disease incidence at the final time point of the experiment (30 dpi, *Figure 2A–B*). To establish a link between phenotypic variation measured in vitro and in vivo, we correlated mutant trait variation with root colonization and plant protection during different phases of the experiment. Trait correlations with the root colonization and plant protection became more significant in time and the most significant correlations were observed at the final time point (30 dpi), followed by middle (15 dpi) and early (5 dpi) sampling time points (*Table 1*). Specifically, swarming motility predicted the root colonization during the seedling stage (5 dpi, *Table 1*; *Figure 2C*, $F_{1,46}$=7.65, $R^2$=0.1239, p=0.0082), while swarming motility was positively associated with plant protection at the flowering stage (30 dpi, *Table 1*; *Figure 2H*, $F_{1,46}$=15.08, $R^2$=0.2306, p<0.001). Similarly, biofilm formation had positive associations with root colonization (*Table 1*; *Figure 2E*, 15 dpi: $F_{1,46}$ = 4.40, $R^2$=0.0675, p=0.0416, 30 dpi: $F_{1,46}$ = 7.62, $R^2$=0.1209, p=0.0089) and plant protection during vegetative and flowering stages (at 15 and 30 dpi), respectively (*Table 1*; *Figure 2J*, 15 dpi: $F_{1,46}$ = 6.44, $R^2$=0.1038, p=0.0146, 30 dpi: $F_{1,46}$ = 8.69, $R^2$=0.1406, p=0.0050). Pathogen suppression was positively associated with root colonization and plant protection at 30 dpi (*Table 1*; *Figure 2F*, $F_{1,46}$=7.65, $R^2$=0.1239, p=0.0082; *Figure 2K*, $F_{1,46}$=15.65, $R^2$=0.2538,

p<0.001), while biomass production was not significantly associated with either root colonization or plant protection at any time points (*Table 1*; *Figure 2D and I*). As a result, the mean performance of mutants ('Monoculture average performance' index based on mean of all traits; see Materials and methods) was significantly positively correlated with both root colonization (*Figure 2G*, $F_{1,46}$=7.77, $R^2$=0.1259, p=0.0077) and plant protection (*Figure 2L*, $F_{1,46}$=28.26, $R^2$=0.3671, p<0.001) at the flowering stage (30 dpi). Together, these results suggest that while mutants with high trait values in biofilm formation, swarming motility, and pathogen suppression had positive effects on root colonization and plant protection, their relative importance varied depending on the growth stage of the plant.

## Designing and testing the performance of mutant consortia in vitro and in vivo

As transposon insertions mainly improved the performance of mutants regarding only one phenotypic trait, we tested if *B. amyloliquefaciens* T-5 performance could be improved by using mutants as phenotypically diverse consortia. To this end, we chose eight mutants that showed improved performance relative to the wild-type strain regarding one of the plant-beneficial traits measured in vitro (*Figure 3—figure supplement 1*, *Supplementary file 2d*; two representative mutants per each measured trait were included). We first tested if these mutants showed negative effects on each other growth in vitro. Based on agar overlay assays, none of the strains clearly inhibited each other in direct contact in co-cultures. Similarly, only slightly negative (up to 12.7%) or positive (up to 7.1%) effects on strains' growth were observed in supernatant exposure experiments (*Figure 3—figure supplement 2*, *Supplementary file 2d*). While direct co-culture experiments are needed to quantify the level of competitiveness between the mutants in the future, this data suggests that transposon insertions made mutants only slightly more inhibitory to each other. Mutants were then used to assemble a total of 37 consortia that varied in their richness level (1, 2, 4, or 8 mutants) and community composition, following a substitutive design where each mutant was equally often present at each richness level (see left panel key of *Figure 3—figure supplement 3* for detailed composition of consortia). We hypothesized that consortia could show improved performance due to phenotypic complementarity or multifunctionality, where different mutants would 'specialize' respective to one of the four phenotypic traits, overcoming trade-offs and potential antagonistic pleiotropy experienced at the individual strain level (*Figure 1B*). We first tested the consortia performance regarding the four traits measured in vitro. We found that relative to wild-type strain, only a few consortia showed improved performance regarding swarming motility (8 of 37), biomass production (2 of 37), biofilm formation (4 of 37), or pathogen suppression (15 of 37) (*Figure 3—figure supplement 3A–D*). Moreover, consortia performance did not show clear relationship with increasing richness regarding to any of the measured traits (*Figure 3—figure supplement 4*). We further tested if the consortia performance could be predicted based on the trait averages of individually grown mutants, assuming that mutant performance is not affected by interactions between the consortia members. Only one significant positive relationship was found between the predicted pathogen suppressiveness, and the size of the inhibition halo observed in vitro lab measurements (*Figure 3A–D*). This suggest that individually measured mutant traits poorly predicted observed consortia performance in vitro except for the pathogen suppression.

We next tested the consortia performance regarding root colonization and plant protection in vivo. While only 7 of 37 of consortia showed improved rhizosphere colonization, around half of them (18 of 37) exhibited a clear increase in plant protection compared to the wild-type strain (at 30 dpi, *Figure 3—figure supplement 3E–F*). While root colonization or plant protection could not be predicted based on the consortia performance measured in vitro (*Figure 3E–F*), increasing consortia richness improved both root colonization (*Figure 3G*, $F_{1,35}$ = 6.52, $R^2$=0.1330, p=0.0152) and plant protection (*Figure 3H*, $F_{1,35}$ = 18.64, $R^2$=0.3289, p<0.001), which were also positively correlated with the consortia average performance measured in vitro (root colonization: $F_{1,35}$ = 6.47, $R^2$=0.1319, p=0.0156; plant protection: $F_{1,35}$ = 8.82, $R^2$=0.1786, p=0.0053; *Figure 3I–J*). We also analyzed the significance of mutant identity effect on the consortia performance in vivo by excluding each strain from the dataset and comparing model fit and significance of explanatory variables. The presence of M54 mutant (efficient in biofilm formation) significantly increased consortia root colonization, while the presence of mutants M59 (efficient in biomass production) and M143 (efficient in biofilm formation) significantly improved plant protection (*Figure 3—figure supplement 5*, *Supplementary file 2e*). Crucially, the effect of consortia richness remained significant after sequential removal of each

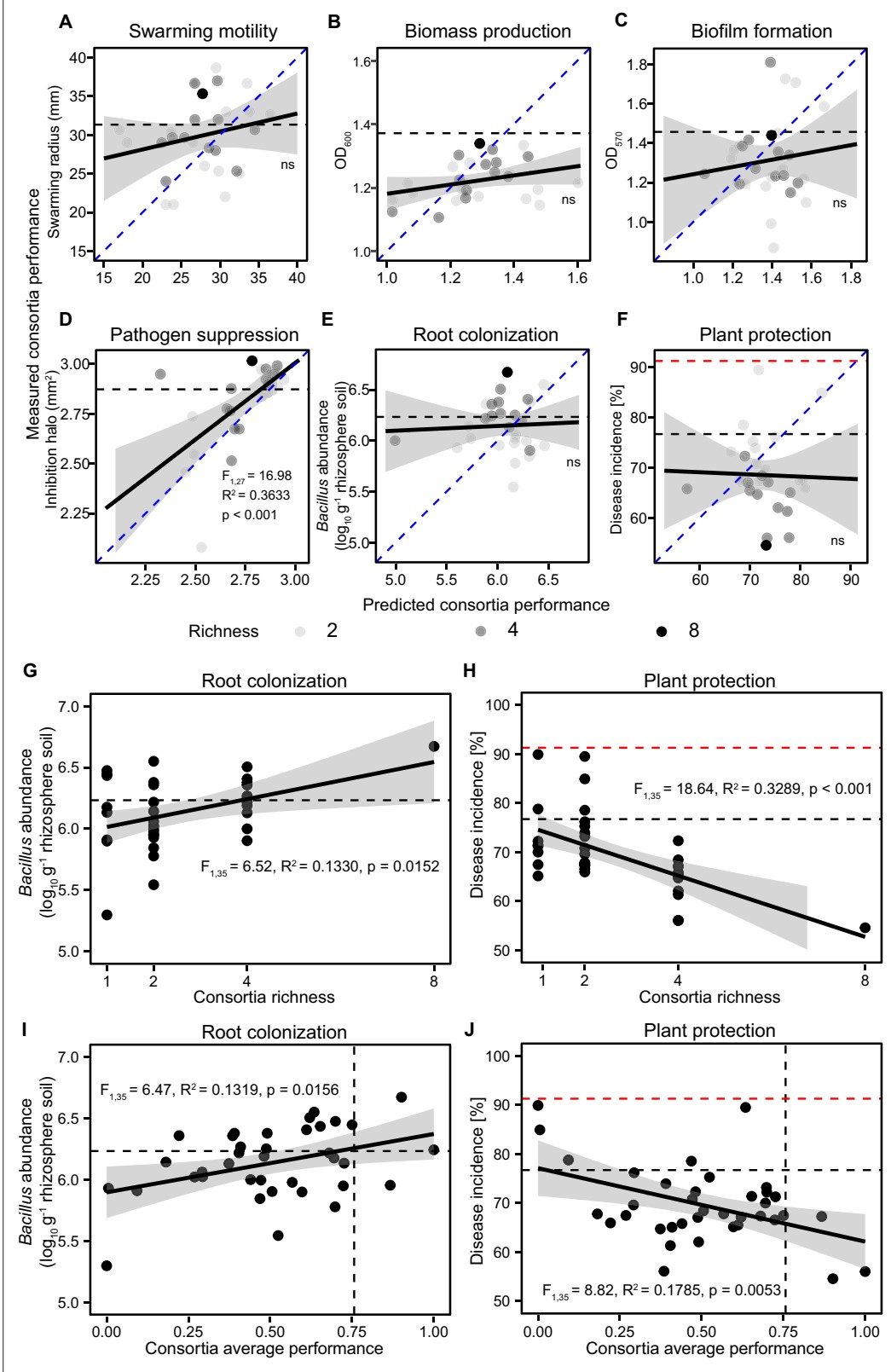

**Figure 3.** The relationships between predicted and observed consortia performance measured in vitro and correlations between plant performance, consortia richness, and consortia average performance measured in vivo. (**A–F**) show correlations between predicted and observed consortia performance regarding swarming motility, biomass production, biofilm formation, pathogen suppression, root colonization, and plant protection, respectively

*Figure 3 continued*

(blue dashed lines show 1:1 theoretical fit and solid black lines show the fitted regression between predicted and observed values). (**G** and **H**) show regression models where root colonization and plant protection were explained by *B. amyloliquefaciens* consortia richness, respectively. (**I** and **J**) show regression models where root colonization and plant protection were explained by *B. amyloliquefaciens* consortia average performance measured in vitro, respectively. In all panels, the black dashed lines show the performance of the wild-type strain, while red dashed lines in (**F**, **H**, and **J**) show the disease incidence of pathogen-only control treatment. In all panels, shaded areas show the confidence interval around the mean.

The online version of this article includes the following figure supplement(s) for figure 3:

**Figure supplement 1.** Phenotypic characteristics of the eight *B. amyloliquefaciens* mutants used for assembly of consortia richness gradient.

**Figure supplement 2.** Pairwise interactions between eight mutants used in diversity-ecosystem functioning experiment based on supernatant culture assays (**A**) and agar overlay spot assays (**B**).

**Figure supplement 3.** The performance of single mutants and mutant consortia relative to wild-type strain regarding four traits measured in vitro and root colonization and plant protection measured in vivo.

**Figure supplement 4.** The relationship between *B. amyloliquefaciens* mutant consortia richness and measured consortia trait performance in vitro.

**Figure supplement 5.** Analysis of mutant identity effects on consortia performance in vivo.

mutant and refitting of the model, which demonstrates that the diversity effect was robust and relatively more important in explaining root colonization and plant protection compared to mutant identity effects (*Supplementary file 2f*). Together, these data suggest that mutant consortia diversity was positively linked with consortia performance in vivo, indicative of positive diversity-ecosystem functioning relationship.

To test if the positive diversity effect was potentially driven by consortium multifunctionality, we compared the performance of the 'optimized' 8-mutant consortium (no. 37; *Figure 3—figure supplement 3*) used in the diversity-ecosystem functioning experiment with eight randomly assembled 8-mutant consortia, which could be considered as phenotypically 'unoptimized'. We found that the 'optimized' consortium was more effective at both plant root colonization and plant protection compared to 'unoptimized' 8-mutant consortia (*Figure 4A and B*). We also correlated the relative performance of all 8-mutant consortia with both in vivo traits and found that both root colonization (*Figure 4C*, $F_{1,7} = 26.56$, $R^2=0.7616$, p=0.0013) and plant protection (*Figure 4D*, $F_{1,7} = 6.39$, $R^2=0.4026$, p=0.0394) improved along with the increase in the relative performance of consortia. Together, these results suggest that in vitro and in vivo phenotyping could reliably predict the root colonization and plant protection of 8-mutant *B. amyloliquefaciens* consortia.

## Discussion

In this work, we tested if increasing intra-species diversity of *B. amyloliquefaciens* T-5 bacterium via mutagenesis could offer a viable strategy for improving mutant consortia multifunctionality and plant health (*Figure 3G–H*). Our results show that mutations that improved bacterial performance regarding one trait often led to specialism and reduced performance regarding other traits. Such trait trade-offs experienced at the individual genotype level could be overcome by assembling consortia of phenotypically distinct mutants, that showed increase in average trait performance. Crucially, the consortia richness and average trait performance correlated positively with increased root colonization and plant protection, indicative of increased consortia multifunctionality and improved plant health. Especially, diverse 8-mutant consortium that consisted of phenotypically distinct mutants performed better compared to randomly assembled 8-mutant consortia consisting of phenotypically similar mutants. Together, these findings suggest that increasing intra-species functional diversity could offer an easy solution for improving the performance of bacterial consortia.

We specifically focused on four *B. amyloliquefaciens* traits that have previously been linked to plant growth promotion: swarming motility, biomass production, biofilm formation, and direct pathogen suppression via antibiosis (*Chen et al., 2013*; *Huang et al., 2020*; *Fira et al., 2018*; *Moreno-Velandia et al., 2019*). While most transposon insertions reduced the strains' performance, some of them improved at least one measured trait. However, all trait improvements were costly and reduced mutants'

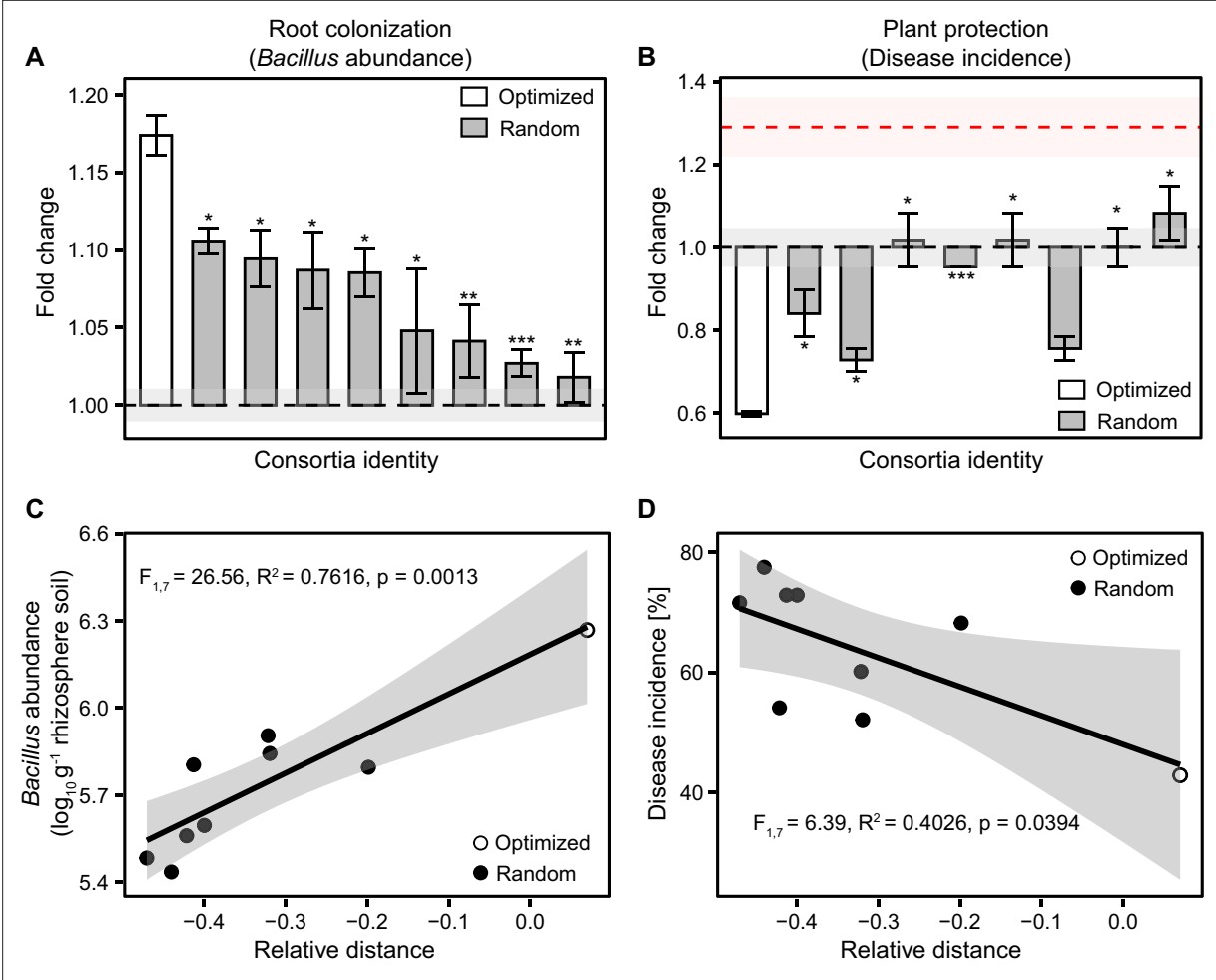

**Figure 4.** Optimized and randomly assembled 8-mutant consortia show contrasting effects on root colonization and plant protection. (**A–B**) compare differences between optimized (white bar) and randomly assembled (gray bars) *B. amyloliquefaciens* 8-mutant consortia on root colonization and plant performance based on Student's t-test (n=3): *** denotes for statistical significance at p<0.001; ** denotes for statistical significance at p<0.01; * denotes for statistical significance at p<0.05. In both (**A** and **B**), Y-axes show the consortia performance as a fold change relative to wild-type strain and shaded areas represent the mean ± SEM. (**C–D**) show positive and negative correlations between consortia relative performance (calculated based on average performance and trait deviance, see Materials and methods) with root colonization (**C**) and disease incidence (**D**); optimized and randomly assembled 8-mutant consortia are shown as white and black circles, respectively (shaded area shows the confidence interval around the mean).

performance regarding other traits, indicative of trade-offs and antagonistic pleiotropy (*Eyre-Walker and Keightley, 2007*). Such costs of adaptation are common with microbes and have previously been linked to a wide range of functions, including metabolism, antibiotic production, motility, and stress resistance (*Yang et al., 2019*; *Ferenci, 2016*). Overall, transposon insertions were identified in several genes associated with broad range of functions (*Figure 1—figure supplement 3*, *Supplementary file 1c*). With eight phenotypically distinct mutants that were used for consortia assembly experiment, increased swarming motility was associated with insertions in *parE* (DNA topoisomerase IV subunit B) and *DeoR* (DNA-binding transcriptional repressor) genes, which has previously been linked to antibiotic resistance and bacterial deoxyribonucleoside and deoxyribose utilization (*Saxild et al., 1996*), respectively. While it remains unclear how these genes were linked with swarming motility, their disruption also affected other traits as evidenced by reduced biofilm formation. Increased biomass production was linked to disruption of *comQ* (competence protein) and *hutI* (imidazolonepropionase) genes and trade-offs with the other three measured traits. *ComQ* gene controls the production of ComX pheromone (*Bacon Schneider et al., 2002*) and has recently been linked to antimicrobial activity (*Esmaeilishirazifard et al., 2018*), which could explain reduction in the pathogen suppression by this mutant. Insertion in histidine utilization (hut) system gene, *hutI*, could have potentially

impaired catabolite and amino acid repression, resulting in improved biomass production with one of the mutants (*Eda et al., 1999*; *Bender, 2012*). Interestingly, increased biofilm formation was also linked to insertion in hut system (*hutU*) with one mutant, while the other mutant had insertion in *YsnB* gene, which encodes for a putative metallophosphoesterase. While both insertions were linked to trade-offs with swarming motility and biomass production, they are not commonly associated with biofilm formation in *Bacillus* (*Dragoš et al., 2018*; *Prágai et al., 2001*). Finally, moderate improvement in pathogen suppression was observed with two mutants that had insertions in *nhaC* (sodium-proton antiporter) and *dfnG* (difficidin polyketide synthesis) genes. The *nhaC* gene is known to act as repressor for Pho regulon in *Bacillus* (*Prágai et al., 2001*), while mutations in Pho regulon have been linked to increased antibiotics production with several *Streptomyces* species (*Santos-Beneit, 2015*). Moreover, the *dfnG* gene controls the production of difficidin antibiotic, which has previously been linked to biocontrol activity against fire blight and *Xanthomonas oryzae* rice pathogen (*Wu et al., 2015*) and could have also suppressed the growth of *R. solanacearum*. Similar to the other phenotypically distinct mutants, insertions associated with increased pathogen suppression led to trade-offs with other measured traits. While more work is required to unequivocally link these mutations with associated traits at the molecular level, our findings show that all above trait improvements achieved via mutagenesis resulted in trade-offs with other traits and that this technique could be used to identify genes underlying plant growth promotion and pathogen suppression.

To overcome trait trade-offs experienced at the individual mutant level, we tested if we could improve the plant growth promotion by combining phenotypically distinct mutants into multifunctional consortia. We found that mutant performance measured in monocultures was a poor predictor of *B. amyloliquefaciens* performance in consortia in vitro, except for the pathogen suppression. This suggest that while the selected mutants did not show direct antagonism toward each other, they likely interacted in other ways leading to unpredictable trait expression when growing together (e.g. via certain emergent effects; *Goldford et al., 2018*). Despite this, we found clear diversity effects, where consortia richness and average performance were positively associated with both plant root colonization and plant protection from *R. solanacearum* pathogen infection. While important mutant identity effects were also observed, omission of these strains did not change the significance of underlying diversity effects, highlighting the importance of interactions between the consortia members in determining the positive effects on the plant health. Together, these results suggest that consortia that performed well regarding all measured traits on average had improved ability to colonize rhizosphere and suppress the pathogen. To test if these effects were driven by diversity per se or underlying trait variation between consortia members, we compared the optimized 8-mutant consortium with eight randomly assembled 8-mutant consortia that were phenotypically more similar. We found that randomly assembled consortia performed less well on average, while consortia functioning improved along with consortia relative performance, suggesting that diverse consortia performed better only when they had been assembled from phenotypically dissimilar mutants. While similar positive diversity-ecosystem functioning relationships have previously been found in more complex *Pseudomonas* (*Hu et al., 2021*; *Hu et al., 2016*; *Hu et al., 2017*), leaf bacterial (*Laforest-Lapointe et al., 2017*) and grass-land soil microbiomes (*Wagg et al., 2019*), we here show that this pattern also holds along with intra-species diversity gradient. While the measured phenotypic traits are considered to be robust to inoculum densities, it will be important to evaluate in the future if the absolute abundances of each mutant play a role in the consortia functioning.

Positive diversity effects have previously been explained by facilitation, ecological complementarity, and division of labor, which can reduce competition between the consortia members within or between niches (*Dragoš et al., 2018*; *Martin et al., 2016*; *Bulleri et al., 2016*). Moreover, high diversity could provide stability for consortia functioning via insurance effects by increasing the likelihood of certain consortia members surviving in the soil after the inoculation (*Yachi and Loreau, 1999*; *Boles et al., 2004*). While our experiments were not designed to disentangle the relative importance of these potential mechanisms, we analyzed which mutant traits could significantly explain the dynamics of root colonization and plant protection at seedling, vegetative, and flowering stages of the tomato growth by focusing on 47 mutants. While the effect of *Bacillus* biomass production was consistently non-significant, both motility and biofilm formation were positively associated with the root colonization. However, motility was significant only at the seedling stage, while biofilm became significant during vegetative and flowering stages. In line with ecological succession often taking place in the

rhizosphere (**Chen et al., 2013**), high motility might have allowed faster colonization of relatively 'sterile' young roots by *Bacillus*, while biofilm formation could have promoted stress tolerance and resource competition in more diverse and mature microbial communities during the later stages of tomato growth (**Bais et al., 2004**; **Xu et al., 2013**). Interestingly, increase in pathogen suppression, biofilm formation, and motility were positively associated with improved plant protection at the flowering stage, which suggests that all these traits were positively associated with the ecosystem functioning in terms of plant health. It is thus possible that consortia were together able to overcome the trait trade-offs experienced at the individual mutant level, leading to improved and more stable ecosystem functioning for the whole duration of tomato growth cycle. In addition, it is possible that some of the measured plant growth promotion traits might act as public goods (**Lee et al., 2010**; **Driscoll et al., 2011**), which could have been shared between different mutants, leading to overall improvement in the mutant population fitness. Such micro-scale mechanisms could be potentially validated in the future using transcriptomics and barcoded Tn-seq mutants, which would allow estimating activity and changes in mutant frequencies during bioinoculation.

In conclusion, we here demonstrate that the beneficial effects provided by a single *B. amyloliquefaciens* bacterium can be improved by increasing consortia functional diversity using transposon mutagenesis. Our approach highlights the importance of intra-species genetic diversity for the ecosystem functioning and provides a trait-based approach for designing microbial communities for biotechnological applications. Our approach does not require a priori knowledge on specific genes or molecular mechanisms, but instead relies on generation of trait variation which is screened and selected by the experimenter. Our method can also help to identify novel functional roles of previously characterized and uncharacterized genes. While the benefit of this method was here demonstrated in the context of agriculture, it could be applied in other biotechnological contexts, such as biofermentation, waste degradation, and food manufacturing. Future work focusing on the population dynamics, metabolism, and gene expression of different mutants will help to understand the relative importance of ecological complementarity, division of labor, insurance effects, population asynchrony (**Blüthgen et al., 2016**), and facilitation for the consortia ecosystem functioning.

## Materials and methods

### Bacterial strains and culture conditions

We used phytopathogenic *R. solanacearum* QL-Rs1115 (**Wei et al., 2011**) and *B. amyloliquefaciens* T-5 biocontrol strains (**Tan et al., 2013**; **Wang et al., 2017**) as our model bacterial species. The *B. amyloliquefaciens* T-5 can suppress the growth of *R. solanacearum* QL-Rs1115 by competing for space and nutrients in the rhizosphere (**Tan et al., 2016**) and by producing various antibacterial secondary metabolites (**Yang et al., 2019**). Both bacterial stocks were cryopreserved at –80°C in 30% glycerol stocks. Prior starting the experiments, active cultures were prepared as follows: *B. amyloliquefaciens* T-5 was grown at 37°C in Lysogeny Broth (LB-Lennox, 10.0 g L$^{-1}$ Tryptone, 5.0 g L$^{-1}$ yeast extract, 5.0 g L$^{-1}$ NaCl, pH = 7.0) and *R. solanacearum* QL-Rs1115 was grown at 30°C in Nutrient Broth (NB, 10.0 g L$^{-1}$ glucose, 5.0 g L$^{-1}$ peptone, 0.5 g L$^{-1}$ yeast extract, 3.0 g L$^{-1}$ beef extract, pH = 7.0) for 24 hr.

### Generation of *B. amyloliquefaciens* T-5 transposon mutant library

To increase the intra-species diversity of *B. amyloliquefaciens*, we generated a random transposon insertion mutant library by using a TnYLB-1 transposon derivative, carried in the thermosensitive shuttle plasmid pMarA (**Supplementary file 2g**), which was electrotransformed to bacteria as previously described by **Zakataeva et al., 2010**; **Ito and Nagane, 2001**. The cells with intact pMarA plasmid contained resistance cassettes to both erythromycin and kanamycin, while the cells with integrated transposons were resistant only to kanamycin. Transposon mutant library was created as follows. An overnight *B. amyloliquefaciens* T-5 cell culture grown in neutral complex medium (NCM, 17.4 g L$^{-1}$ K$_2$HPO$_4$, 11.6 g L$^{-1}$ NaCl, 5 g L$^{-1}$ glucose, 5 g L$^{-1}$ tryptone, 1 g L$^{-1}$ yeast extract, 0.3 g L$^{-1}$ trisodium citrate, 0.05 g L$^{-1}$ MgSO$_4$.7H$_2$O, and 91.1 g L$^{-1}$ sorbitol, pH = 7.2) was diluted 25-fold with fresh NCM supplemented with 5 mg mL$^{-1}$ of glycine and grown at 30°C for 3 hr on a rotary shaker (170 rpm). After 1 hr incubation (at an optical density OD$_{600}$~ 0.8), cells were cooled on ice, harvested by centrifugation (8000×*g* for 6 min at 4°C) and washed four times with ice-cold electrotransformation buffer (ETM, 0.5 M sorbitol, 0.5 M mannitol, and 10% glycerol). Resulting pellets were resuspended in ETM

buffer supplemented with 10% PEG 6000 and 1 mM $MgCl_2$, yielding approximately $10^{10}$ cells $mL^{-1}$. Cells were then mixed with 500 ng of plasmid DNA in an ice-cold electrotransformation cuvette (2 mm electrode gap), and after 1–3 min incubation at room temperature, exposed to a single electrical pulse using a MicroPulser Electroporator (Bio-Rad Laboratories) at field strength of 7.5 kV $cm^{-1}$ for 4.5–6 ms. Immediately after the electrical discharge, cells were transferred into 1 mL of LB, incubated with gentle shaking at 30°C for 3–8 hr, and plated on LB agar containing 10 µg $mL^{-1}$ erythromycin. Transformants were selected after 36–48 hr incubation at 30°C. To generate final transposon library, erythromycin-resistant colonies with plasmids were individually transferred to fresh LB and incubated overnight at 30°C, after cultures were diluted, spread on LB plates supplemented with 10 µg $mL^{-1}$ kanamycin, and incubated for 24 hr at 46°C. As the plasmid cannot replicate at 46°C, only cells with an integrated transposons grew and could be separated. A total of 1999 transformed colonies were isolated and individually cryopreserved in 30% glycerol at –80°C.

## Phenotypic characterization of *B. amyloliquefaciens* T-5 mutant library in vitro

The wild-type strain and 1999 mutants were phenotyped for following plant-growth promoting traits: swarming motility, biomass production, biofilm formation, and pathogen suppression via production of antimicrobials (see below). These traits were selected due to their known importance for *B. amyloliquefaciens* competitiveness in the rhizosphere and their involvement in pathogen suppression (*Chen et al., 2013*; *Huang et al., 2020*; *Fira et al., 2018*; *Moreno-Velandia et al., 2019*). To prepare bacterial inoculants, frozen colonies were picked and pre-grown overnight in LB at 37°C, washed three times in 0.85% NaCl and adjusted to initial $OD_{600}$ of 0.5 (~ $10^7$ cells $mL^{-1}$, based on OD vs colony forming unit [CFU] calibration curve, *Figure 1—figure supplement 4*). In addition to each individual trait, we also calculated the average of all measured traits and used the resulting 'monoculture average performance' (*Wagg et al., 2014*) index to compare mutants' overall performance. Of the 1999 phenotyped mutants, a subset of 479 mutants were randomly selected for more detailed analysis and probiotic bioinoculant design. While we likely missed certain mutants with this method, the 479 mutants represented a similar phenotypic diversity as the 1999 mutant collection (Mantel test; r=0.7591, p=0.04167), indicating that our sampling captured a phenotypically representative subsample of mutants (*Supplementary file 1a and b*).

*Swarming motility* was measured using a previous method described by Kearns (*Kearns et al., 2004*). Briefly, 2 µL of each *B. amyloliquefaciens* mutant was inoculated into the center of 0.7% agar LB plates supplemented with 10 µg $mL^{-1}$ of kanamycin. After 24 hr incubation at 30°C, swarming motility was evaluated as the radius of the colony. Three replicates were used for each mutant.

*Biomass production* was assessed on 96-wells microtiter plates (at 30°C with agitation) in 200 µL of 'recomposed exudate' medium (abbreviated as 'RE', which contained: 0.5 g $L^{-1}$ $MgSO_4.7H_2O$, 1.0 g $L^{-1}$ $K_2HPO_4$, 0.5 g $L^{-1}$ KCl, 1.0 g $L^{-1}$ yeast extract, 1.2 mg $L^{-1}$ $Fe_2(SO_4)_3$, 0.4 mg $L^{-1}$ $MnSO_4$, 1.6 mg $L^{-1}$ $CuSO_4$, 2 g $L^{-1}$ $(NH_4)_2SO_4$, 0.8 g $L^{-1}$ glucose, 1.3 g $L^{-1}$ fructose, 0.2 g $L^{-1}$ maltose, 0.02 g $L^{-1}$ ribose, 5.6 g $L^{-1}$ citrate, 1.4 g $L^{-1}$ succinate, 0.2 g $L^{-1}$ malate, 0.8 g $L^{-1}$ casamino acids *Nihorimbere et al., 2012*). After 24 hr of growth, $OD_{600}$ was measured as a proxy for biomass production (*Figure 1—figure supplement 4*). Three replicates were used for each mutant.

*Biofilm formation* was assessed as described previously (*Hamon and Lazazzera, 2001*) using 24-well polyvinyl chloride microtiter plates instead of 96-well plates. Briefly, 10 µL *B. amyloliquefaciens* cells were added into 1 mL of biofilm-promoting growth medium (MSgg minimal medium: 2.5 mM PBS [pH 7.0], 100 mM MOPS [pH 7.0], 50 µM $FeCl_3$, 2 mM $MgCl_2$, 50 µM $MnCl_2$, 1 µM $ZnCl_2$, 2 µM thiamine, 50 mg phenylalanine, 0.5% glycerol, 0.5% glutamate, and 0.7 mM $CaCl_2$) on 24-well microtiter plates and incubated without agitation for 24 hr at 30°C (*Branda et al., 2001*). The growth medium and planktonic cells were removed, and remaining cells adhered on well walls were stained with 1% crystal violet dissolved in washing buffer (0.15 M $(NH_4)_2SO_4$, 100 mM $K_2HPO_4$ [pH 7], 34 mM sodium citrate, and 1 mM $MgSO_4$) for 20 min at room temperature. Plates were then rinsed with demineralized water to remove excess crystal violet, after the remaining crystal violet bound to well wall biofilms were solubilized in 200 µL of solvent (80% ethanol, 20% acetone). Biofilm formation was defined as the optical density of crystal violet at $OD_{570}$. Three replicates were used for each mutant.

*Pathogen suppression* via production of antibiotics was assessed as inhibition of *R. solanacearum* QL-Rs1115 strain using an agar overlay assay (*Parret et al., 2005*). Briefly, small volume drops (2 µL) of

each *B. amyloliquefaciens* mutant and wild-type strain were spotted on NA soft agar plates and incubated for 24 hr at 30°C. Next, these plates were chloroform-fumigated to kill all the bacteria (*Parret et al., 2005*), leaving only the secreted antimicrobials and then fully covered with *R. solanacearum* suspension (with a final concentration of approximately $10^7$ cells mL$^{-1}$). The pathogen suppression of each mutant was defined as the area of *R. solanacearum* inhibition halo around the *B. amyloliquefaciens* colony (in mm$^2$), which is proportional to antibiotic production (*Delignette-Muller and Flandrois, 1994*). Three replicates were used for each mutant.

## Selecting a representative subset of *B. amyloliquefaciens* T-5 mutants for greenhouse experiments

In order to select a representative subset of mutants for greenhouse experiments, we first used K-means clustering (*Hartigan and Wong, 1979*) to divide the wild-type and 479 phenotyped mutants into clusters based on swarming motility, biomass production, biofilm formation, and pathogen suppression (*Supplementary file 1b*). Briefly, K-means clustering assigns n observations into k clusters where each observation (in our case mutant) belongs to a cluster with the nearest mean (cluster centers or cluster centroid). According to the gap statistic method (*Tibshirani et al., 2001*), three clusters was the optimum number (k) for this dataset (*Figure 1—figure supplement 2*). With this method, each mutant was hence assigned to one of the clusters. Clustering was further visualized using principal component analysis (PCA) based on the first two principal components to show the variety of mutants. We randomly selected approximately 10% of strains from each cluster, resulting in a subset of 47 mutants, which were used for greenhouse experiments (26, 11, and 10 mutants from clusters 1, 2, and 3, respectively, *Supplementary file 1c*). These 47 mutants were further analyzed to determine the disrupted genes by TnYLB-1 transposon insertion using the inverse polymerase chain reaction (IPCR) method as previously described by *Le Breton et al., 2006*. First, 5 µg of genomic DNA isolated from each respective transposon mutant was digested with Taq I and circularized using 'Rapid Ligation' kit (Fermentas, Germany). IPCR was carried out with ligated DNA (100 ng), using oIPCR1 and oIPCR2 primers (*Supplementary file 2h*). The cloned sequences were then purified using PCR purification kit (Axygen, UK) and the flanking genomic regions surrounding the transposon insertion sites were sequenced using the primer oIPCR3 (*Supplementary file 2h*). Obtained DNA sequences were compared against available databases (GenBank and *Bacillus* Genome Data-base) using the BLASTX and BLASTN available at the NCBI, and against the complete ancestral *B. amyloliquefaciens* T-5 genome sequence (Accession: CP061168, *Figure 1—figure supplement 3A*, *Supplementary file 1c*). The functional classification of disrupted genes for all 47 transposon mutants is summarized in *Figure 1—figure supplement 3B*.

## Assessing the performance of individual *B. amyloliquefaciens* T-5 mutants in a greenhouse experiment

All selected 47 mutants and the wild-type strain were individually screened for their ability to colonize tomato rhizosphere and protect plants against infection by *R. solanacearum* QL-Rs1115 pathogen strain in a 50-day-long greenhouse experiment. Surface-sterilized tomato seeds (*Lycopersicum esculentum, cultivar 'Jiangshu'*) were germinated on water agar plates in the dark at 28°C for 2 days, before sowing to sterile pots containing wet vermiculite (Huainong, Huaian Soil and Fertilizer Institute, Huaian, China). Ten-day-old tomato seedlings (at three-leaves stage) were then transplanted to seedling trays containing natural, non-sterile soil collected from a tomato field in Qilin Town, Nanjing, China (*Chen et al., 2013*). Plants were inoculated with individual *B. amyloliquefaciens* mutants by drenching, resulting in a final concentration of $10^7$ CFU g$^{-1}$ soil (*Wei et al., 2013*). The *R. solanacearum* strain was inoculated using the same method 1 week later at a final concentration of $10^6$ CFU g$^{-1}$ soil. Positive control plants were treated only with *R. solanacearum*, while negative control plants received no bacterial inoculants. Three replicated trays were set up for each treatment, with 20 seedlings (in individual cells) per tray. Each tray was considered as one biological replicate. Tomato plants were grown for 30 dpi with natural temperature (ranging from 25°C to 35°C) and lighting variation (around 16 hr of light and 8 hr of dark). Seedling trays were rearranged randomly every second day and regularly watered with sterile water.

## Quantifying *B. amyloliquefaciens* mutants' root colonization and plant protection in the rhizosphere

The root colonization and plant protection of 47 *B. amyloliquefaciens* T-5 mutants was quantified individually as a change in their population densities in the tomato rhizosphere after 5, 15, and 30 days of *R. solanacearum* pathogen inoculation ('dpi'). At each sampling time point, three independent plants per inoculated mutant were randomly selected and sampled destructively by carefully uprooting the plant and gently removing the soil from the root system by shaking. After determining plant fresh weight, the root system of each plant was thoroughly ground in 5 mL of 10 mM $MgSO_4 \cdot 7H_2O$ using a mortar, and serial dilutions of root macerates were plated on a semi-selective *Bacillus* medium consisting of 326 mL $L^{-1}$ vegetable juice (V8, Campbell Soup Co., USA), 33 g $L^{-1}$ NaCl, 0.8 g $L^{-1}$ dextrose, 16 g $L^{-1}$ agar (pH 5.2, adjusted with NaOH) supplemented with 45 mg $L^{-1}$ cycloheximide and 22.5 mg $L^{-1}$ polymyxin B (*Kinsella et al., 2009*). This media was used to count the densities of *B. amyloliquefaciens* T-5 wild-type, and the same media supplemented with 10 µg $mL^{-1}$ kanamycin was used to count the densities of mutant strains. Plates were incubated at 30°C for 30 hr and bacterial densities expressed as CFU per gram of root biomass. The effect of *B. amyloliquefaciens* wild-type and mutants on plant protection was measured as the reduction of bacterial wilt disease symptoms during the experiment (based on the proportion of plants showing wilting symptoms). The first wilting symptoms appeared 7 dpi and the proportion of diseased plants quantified at 5, 15, and 30 dpi were used in analyses. Plant protection was expressed as the relative reduction in the number of wilted plants compared to the positive control (only *R. solanacearum* inoculated in the absence of *B. amyloliquefaciens* T-5 mutants or wild-type).

## Assembly of phenotypically dissimilar *B. amyloliquefaciens* mutant consortia

To test if the performance of *B. amyloliquefaciens* T-5 mutants could be improved by using consortia of phenotypically dissimilar mutants, a subset of eight best-performing mutants excelling at different phenotypic traits were selected (*Figure 3—figure supplement 1*, *Supplementary file 2c*). Specifically, these included two mutants that showed high swarming motility (M108: *pare* and M124: *DeoR*), high biomass production (M59: *comQ* and M109: *hutI)*, high biofilm formation (M54: *hutU* and M143: *YsnB*), and slightly improved pathogen suppression (M38: *nhaC* and M78: *dfnG*) relative to the wild-type strain (*Figure 3—figure supplement 1*, *Supplementary file 2c*). To test the effect of transposon insertions on potential antagonism between the mutants, we conducted two types of assays: direct growth inhibition by (1) spotting each strain on top of the others using agar overlays and(2) growing each strain in the supernatant of the other strains. With agar overlay assays, 2 µL of each mutant with density $OD_{600}$ of 0.5 (~$10^7$ cells $mL^{-1}$) was spotted on the soft agar overlay of the other mutants and direct antagonistic effect was measured as the size of the inhibition halo observed on the soft agar plates (*Fields et al., 2022*). For the supernatant assay, we first cultured each mutant in liquid LB for 2 days and collected supernatants by using 0.22 µm filters. In the growth assays, 2 µL of each strain with initial concentration of $10^7$ cells $mL^{-1}$ was mixed with 20 µL of each supernatant and 178 µL of 50% LB. The growth of each strain was measured after 24 hr as optical density ($OD_{600}$), and inhibition calculated as the relative growth of each strain in its own or other strains' supernatant compared to strains' growth in the fresh 50% LB (diluted with sterile water). Here, the reduced growth in other strains' supernatant relative to the growth in the fresh medium was deemed as inhibition between mutants. A following formula was used where $OD_{600\,sup}$ and $OD_{600\,LB}$ denote for mutants' growth in other mutants' supernatant or in 50% fresh LB after 24 hr:

$$Relative\ growth = \frac{OD_{600sup} - OD_{600LB}}{OD_{600LB}} \times 100\%$$

These eight mutants were then used to assemble a total of 29 consortia with 2, 4, or 8 mutants, following a substitutive design where each mutant was equally often present at each community richness level (see left panel key of *Figure 3—figure supplement 3* for detailed consortia assembly). Mutants were mixed in equal proportions in each consortium with final total bacterial density $OD_{600}$ of 0.5 (e.g., 50:50% or 25:25:25:25% in two and four mutant consortia, respectively; ~ $10^7$ cells $mL^{-1}$). This design has previously been used to investigate biodiversity-ecosystem functioning relationships in plant-associated bacterial communities (*Hu et al., 2016*; *Becker et al., 2012*), allowing disentangling

the effects due to consortia richness, composition, and mutant strain identity. In addition, to compare the performance of optimized 8-member consortium (assembled based on phenotypic dissimilarity; see above) with non-optimized 8-mutant consortia, we assembled eight additional 8-mutant consortia randomly from the 479 mutant collection, which were used in in vitro lab and in vivo greenhouse experiments.

## Phenotypic characterization of *B. amyloliquefaciens* consortia performance in vitro and consortia root colonization and plant protection in the tomato rhizosphere

The performance of each mutant and assembled consortium was assessed in vitro in the lab by measuring traits as mono- and co-cultures following the same methods as described previously (swarming motility, biomass production, biofilm formation, and pathogen suppression). Mutant strains were prepared individually from frozen stocks by growing overnight in liquid LB, pelleted by centrifugation ($4000 \times g$, 3 min), washed three times with 0.85% NaCl and adjusted to $OD_{600}$ of 0.5 ($10^7$ cells mL$^{-1}$). Consortia were then assembled following the substitutive design describe earlier (*Figure 3—figure supplement 3*) by mixing mutants in equal proportions for each consortium with total bacterial density $OD_{600}$ of 0.5 ($10^7$ cells mL$^{-1}$; e.g., 50:50% or 25:25:25:25% in two and four mutant communities, respectively). Consortia traits were characterized as described previously and compared with the ancestral *B. amyloliquefaciens* wild-type strain. The root colonization and plant protection of *B. amyloliquefaciens* T-5 consortia were quantified in greenhouse experiments following previously described methods. Predicted performances were calculated following the additive model, equaling the sum of different trait values of each member divided by the richness value of the given consortium. To link the performance of single mutant with functioning of consortia, we used the relative performance measure, which included the magnitude and direction of difference relative to the wild-type strain. The difference and direction in magnitude to the wild-type strain were calculated based on the Euclidean distance and average performance using following formula:

$$Relative\ performance = \frac{\sum_{i=1}^{n} \left( D_i \times \frac{AP_i - AP_{wt}}{|AP_i - AP_{wt}|} \right)}{n},$$

$D_i$, Euclidean distance between each consortium member and wild-type based on four traits; $AP_i$, average performance of each community member; $AP_{wt}$, average performance of wild-type; n, community richness.

## Statistical analyses

Data were analyzed with a combination of analysis of variance (ANOVA), PCA, linear regression models, unpaired two-sample Wilcoxon tests, and Student's t-test. Individually measured mutant traits data was normalized between 0 and 1 across the all collection using min-max normalization (*Jain et al., 2005*). In addition, the different phenotypic traits were combined into a 'Monoculture average performance' index, which was calculated as the mean of the four standardized traits for each mutant. Monoculture average performance and consortia traits values were also min-max normalized between 0 and 1 for subsequent analyses. To classify mutants into different functional groups, K-means clustering algorithm ('fviz_nbclust' in 'factoextra' package and 'kmeans' function) was used and clusters were visualized using PCA ('princomp' in 'vegan' package) based on multivariate trait data. The phenotypic dissimilarity between the same mutants and the wild-type strain was calculated using 'vegdist' based on 'Euclidean' algorithm. The *B. amyloliquefaciens* T-5 abundance data measured in root colonization assays were $log_{10}$-transformed and disease incidence data were arcsine square root-transformed prior the analyses. Linear regression models were used to explain root colonization and plant protection with mutant traits, average performance, consortia richness, and consortia relative performance. Treatment mean differences were analyzed using two-sample Wilcoxon test ('wilcox.test' function) or Student's t-test ('t.test' function) depending on the unequal or equal sample sizes, respectively. The temporal effects of four traits on root colonization and plant protection were assessed separately for different time points using ANOVA ('aov' function). All statistical analyses were

performed using R 3.5.2 (R core Development Team, Vienna, Austria). All code used in this study is available on request from corresponding authors.

## Acknowledgements

We thank Daniel Rozen, Benoit Stenuit, and Zheren Zhang for valuable comments on the manuscript. This research was supported by the National Key Research and Development Program of China (2021YFD1900100, 2022YFF1001804, 2022YFD1500202), the National Natural Science Foundation of China (42325704, 42090064, 41922053, and 31972504), the Fundamental Research Funds for the Central Universities (KYT2023001). V-PF is supported by the Royal Society Research Grants (RSG\R1\180213 and CHL\R1\180031) and jointly by a grant from UKRI, Defra, and the Scottish Government, under the Strategic Priorities Fund Plant Bacterial Diseases programme (BB/T010606/1).

## Additional information

### Funding

| Funder | Grant reference number | Author |
| --- | --- | --- |
| National Key Research and Development Program of China | 2021YFD1900100 | Zhong Wei |
| National Key Research and Development Program of China | 2022YFF1001804 | Xiaofang Wang |
| National Key Research and Development Program of China | 2022YFD1500202 | Tianjie Yang |
| National Natural Science Foundation of China | 42325704 | Zhong Wei |
| National Natural Science Foundation of China | 42090064 | Qirong Shen |
| National Natural Science Foundation of China | 41922053 | Zhong Wei |
| National Natural Science Foundation of China | 31972504 | Yangchun Xu |
| Fundamental Research Funds for the Central Universities | KYT2023001 | Zhong Wei |
| Royal Society Research Grants | RSG\R1\180213 | Ville-Petri Friman |
| Royal Society Research Grants | CHL\R1\180031 | Ville-Petri Friman |
| Strategic Priorities Fund Plant Bacterial Diseases programme | BB/T010606/1 | Ville-Petri Friman |

The funders had no role in study design, data collection and interpretation, or the decision to submit the work for publication.

### Author contributions

Jingxuan Li, Chunlan Yang, Formal analysis, Investigation, Visualization, Writing – original draft; Alexandre Jousset, Conceptualization, Writing – review and editing; Keming Yang, Formal analysis, Investigation, Visualization; Xiaofang Wang, Tianjie Yang, Formal analysis, Investigation; Zhihui Xu, Writing – review and editing; Xinlan Mei, Investigation; Zengtao Zhong, Resources, Methodology, Writing – review and editing; Yangchun Xu, Conceptualization, Supervision, Project administration; Qirong Shen, Supervision, Project administration; Ville-Petri Friman, Conceptualization, Formal analysis,

Visualization, Writing – review and editing; Zhong Wei, Conceptualization, Data curation, Supervision, Funding acquisition, Project administration

### Author ORCIDs
Zhihui Xu (iD) http://orcid.org/0000-0002-3987-8836
Ville-Petri Friman (iD) http://orcid.org/0000-0002-1592-157X
Zhong Wei (iD) http://orcid.org/0000-0002-7967-4897

### Decision letter and Author response
Decision letter https://doi.org/10.7554/eLife.90726.sa1
Author response https://doi.org/10.7554/eLife.90726.sa2

## Additional files

### Supplementary files
• Supplementary file 1. Phenotypic trait data and genetic information of used B. amyloliquefaciens T-5 transposon insertion mutants. (a) The profile of the 1999 mutants and the wild-type T-5. (b) The profile of the 479 mutants and the wild-type T-5. (c) The profile of the selected mutants used in the single mutant greenhouse experiment.

• Supplementary file 2. Supplementary tables for this paper. (a) Comparison of in vitro traits between mutants belonging to three clusters using unpaired two-samples Wilcoxon test. Significant effects ($p<0.05$) are highlighted in bold. (b) Comparison of root colonization and plant protection between mutants belonging to three clusters using unpaired two-samples Wilcoxon test. Significant effects ($p<0.05$) are highlighted in bold and dpi denotes for days post-pathogen inoculation. (c) Comparison of trait values of eight mutants used for the assembly of consortia richness gradient relative to the wild-type strain based on Student's t-test. Significant differences are shown in bold and arrows show increase (upward) and decrease (downward) in trait values. (d) p-Values for comparing the biomass production of each mutant strain on its own or other strains' supernatant compared to fresh 50% LB based on Student's t-test. Significant differences are shown in bold and arrows show facilitative (upward) and antagonistic (downward) interactions between the mutants. The magnitude of these interactions is shown in *Figure 3—figure supplement 1* as a heatmap. (e) Comparison of mutant identity effects on consortia root colonization and plant protection based on the absence and presence of each mutant in consortia. Significant effects ($p<0.05$) are highlighted in bold based on unpaired two-samples Wilcoxon test. (f) Comparison of the mutant identity effects and consortia richness on root colonization and plant protection. Richness was fitted sequentially after mutant identity effects (presence or absence in consortia). Both response variables were treated as continuous variables and *Bacillus* abundance data was log-transformed before the analysis. Significant effects ($p<0.05$) are highlighted in bold. (g) Bacterial strains and plasmid used in this study. (h) Primers used in this study.

• MDAR checklist

• Source data 1. Source data used in this paper.

### Data availability
All data generated or analysed during this study are included in the manuscript and supporting file; Source Data files have been provided in Source data 1.

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
