## [Editor Report]

The work is significant because it reports that intra-species phenotypic diversity in bacteria could be an effective way to improve probiotic consortia in agriculture. Two solid conclusions emerge from this important work: (1) Communities of near-clonal bacteria can be optimized based on functional traits to promote functional diversity. (2) Consortium multifunctionality – but not richness – is key to promoting bacterial colonization in roots and to protection against a plant pathogen. The work has broad practical implications for designing probiotic consortia that promote host health beyond the plant-microbe interaction field.

---

## [Decision Letter]

**Decision letter after peer review:**

[Editors’ note: the authors submitted for reconsideration following the decision after peer review. What follows is the decision letter after the first round of review.]

Thank you for submitting the paper "Engineering ecologically complementary rhizosphere probiotics using consortia of specialized bacterial mutants" for consideration by *eLife*. Your article has been reviewed by 3 peer reviewers, one of whom is a member of our Board of Reviewing Editors, and the evaluation has been overseen by a Senior Editor. The following individual involved in the review of your submission has agreed to reveal their identity: Jacob D Palmer (Reviewer #2).

Comments to the Authors:

We are sorry to say that, after consultation with the reviewers, we have decided that this work will not be considered further for publication by *eLife*. Note that while we cannot publish the current study, we remain in principle interested in the work. If you are able to seriously revise the work and respond to the concerns, we would be prepared to review the work again, although we would treat it as a new submission.

As you will see, although the reviewers see the work as novel and with the potential to design probiotic consortia, they have also raised a number of issues that will require new experiments, new analyses and important reformatting of the manuscript. After internal discussions with the reviewers, we remain in principle interested in the work but we felt that it is important to give you more time if you wish to thoroughly address the concerns. Particularly, performing pairwise interactions among mutants, testing random sets of 8-member consortia, and reorganizing the manuscript appears to be critical to corroborate the claims and improve clarity, respectively. Furthermore, the reviewers think that the experiments were not designed to test ecological theories regarding ecosystem functioning and stability but rather as a method to design probiotic consortia in agriculture. Many concepts mentioned in the introduction and discussion might not be relevant here and might distract rather than inform.

*Reviewer #1 (Recommendations for the authors):*

Overall, the concept/methodology is interesting and the results open novel insights towards the engineering of microbial consortia that promote disease resistance. Furthermore, the statistical analyses seem to be conducted and represented adequately. The text is well written and methods are nicely detailed. Despite the novelty, the manuscript has some, yet important weaknesses that will need to be addressed.

Strengths:

Multi-species inoculants often fail to promote plant health under natural conditions because microbe-microbe interactions are not often considered when consortia are designed and assembled. Therefore, the idea of promoting intra-species multi-functionality through the assembly of bacterial mutants derived from a single strain is exciting. This is because it could promote functional diversity whilst at the same time minimizing the risk of competition between strains. The research question is therefore interesting and novel, the combination of methods, which include strain engineering (Tn5 mutant library), large-scale phenotypic screening and assembly of synthetic microbial consortia is adequate to test the relevance of functional complementary and multi-functionality between mutant strains.

The greenhouse experiment using a natural soil is also relevant to test the effects of the inoculants under semi-natural conditions.

The statistical analysis is adequate, the manuscript well written and the figures have been nicely crafted.

The conclusions are largely supported by experimental evidence, although it remains difficult to unambiguously conclude that the method actually works.

Weaknesses: There are some weaknesses.

First, the authors used an initial random pool of 479 mutants for in vitro trait phenotyping. It remains unclear why more mutants have not been tested given the fact that these are high-throughput in vitro screens. The authors did not use a saturated mutant library and might have overlooked many important genes.

Second, the structure of the manuscript is confusing. It remains unclear how the three clusters in Figure 1 have been defined and why they were not considered for the rational selection of the consortia shown in Figure 2. The most confusing part comes from the last paragraph and Figure 3 which shows data based on individual strains and not consortia. This is redundant with the data shown in Figure 1 and this part provides very limited information regarding the "underlying mechanisms between consortia diversity and improved performance". As it is presented this part distracts rather than informs. I do not think that this is the rights way to test the "insurance hypothesis". this hypothesis makes sense if tested in a community context with consortia of different complexities (as in Figure 2).

It remains also unclear whether complementarity between specialized bacterial mutants is really needed for improving plant performance (as stated in the title). The authors showed that the rational selection based on Figure 1 has some limits to predicting community behavior in vitro and in planta. Therefore, a key question is whether microbial consortia composed of random pools of 8 mutants (i.e. not pre-selected based on in vitro phenotypes) would provide similar fitness benefits as the mutant pool selected in Figure 2. In other words, is the first part of the paper really required or could one simply inoculate a pool of Tn5 mutants and observe the same protective effect? Testing the effects of several independent mutant consortia would be needed to draw general conclusions.

Finally, the paper is mechanistically weak. That being said, this is not the core of the manuscript and therefore it is not an issue. Although requesting complementation lines for the 8 mutants used for consortia establishment is beyond the scope of the manuscript, it is important to keep in mind that it is difficult to conclude anything regarding the bacteria genetic determinants identified here without proper functional validations.

To strengthen the conclusions shown in Figure 2, it would be important to test other independent consortia of 8 mutants and validate directionality in their effects depending on whether they were selected randomly or based on rational criteria. The concern is that part 1 and part 2 of the manuscripts might not be connected and it would be important to validate that the rational selection based on in vitro phenotypes is actually required and important.

The results presented in Figure 3 (individual strains) are insufficient to test the insurance effect. One would need to inspect this in the context of the various consortia shown in Figure 2. Here, it would be important to assess 1. total Bacillus load (e.g. using colony counts), 2. disease suppression and 3. "community structure" of the bacilli mutants. The latter can likely easily be conducted using PCRs with pairs of Tn5 transposon specific and random primers similar to classical TnSeq approaches. If the insurance hypothesis holds true, one would expect, that throughout the whole growth cycle, the complex community always performs better than the individual mutants (based on colony counts and disease suppressiveness) and also that relative abundances of Bacilli mutants change across the growth cycle. Such experiments could further reinforce the conclusion regarding insurance effects. Alternatively, authors should consider including this section as Figure 2 and tone-down their claim regarding these insurance effects that are not corroborated by experimental evidence.

Agar overlay data not shown?

*Reviewer #2 (Recommendations for the authors):*

This study provides a novel, potentially high-throughput method for identifying beneficial microbial consortia towards plant growth and health, which may be applicable towards other biotechnological applications. This study has the potential to be of great interest to all those working in host-microbe interactions. However, it is my opinion that there is an overemphasis on microbial ecology and the inherent value of intraspecies diversity towards ecosystem health, where I think the data of this study does not meet their intended aims.

Strengths: This study provides a novel and unique approach for studying the impact of intraspecies diversity on microbial consortia and understanding host (plant)-microbe interactions. There is a very good introductory section highlighting the field of microbial ecology. The methods used seem applicable to many different potential environments, as highlighted by the authors, and might be of great interest to those working on both applied and fundamental areas of microbiology and/or host-microbe interactions. The justification of the study, the introduction, and the methods used are all very high quality. The stated expectations by the authors are also well-supported by the theoretical literature and clearly explained. The focus on intraspecies interactions and diversity is also very welcomed and appears to me to be a novel approach.

Weaknesses: It is my opinion that the narrative in the results and discussion is overly focused on the impact of intraspecies diversity in microbial ecology, and how this diversity improves ecosystem functioning. I do not think the study provides support for these claims. Instead, the study does provide excellent support for high-performing consortia of mutants, which outperform the WT monoculture across varying metrics, which has both applied and fundamental value regarding agriculture and host-microbe interactions at both the strain and molecular levels. The data provided as apart of this work provides many new avenues of research and I think is an excellent step towards developing beneficial probiotic consortia. Unfortunately, these advances are sometimes overshadowed by claims of ecosystem stability and ecosystem functioning that lack support in the data. In multiple places in the text, it is suggested that intraspecies consortia are not subject to intra-species conflict. I find this to be potentially misleading, as it implies that these genotypes now have neutral (or positive) interactions, which is not appropriately measured nor expected from the data.

Line 44-47 – I am unsure of the support that antagonistic traits often increase invasion success of probiotic bacteria, or what this statement really means. I think it could benefit by being more explicit.

The introduction may benefit from a more thorough explanation of biodiversity-ecosystem functioning theory (line 84), to match the excellent introduction of microbial ecology from lines 50-75.

Line 95: 'benefits' to 'beneficial

Line 96: Expected advances. It does not appear that expectations 2 or 3 are tested. Expectation 3 is also rather unclear. Please consider rewriting this section to better set the reader's expectations for the data and results in the paper.

Line 162 – Please clarify how it is known that OD600 = 0.05 equals ~10^5 cells/mL for B. amyloliquifaciens. Or omit the ~10^5 cells/mL. Additionally, I wonder if OD600 is an appropriate proxy for biomass production in a B. amyloliquifaciens mutant library? Plating and counting CFU/mL would be the much more reliable measurement. Considering the method used for biomass production measurement, did you perform these measurements in a plate reader and have additional time series data beyond the 24hr timepoint? Perhaps maximum growth rate or other factors of the growth curve may also be good indicators of beneficial phenotypes for plant health.

General Methods Comments: Given the more applied aims of this study, maintaining a constant inoculum size and measuring the outputs of the consortium is a very reasonable strategy. However, performing experiments in this way ignores the intraspecies competition that likely defines the system. This makes claims of facilitation, exploitation, or cooperation moot, as they are not being appropriately measured. The authors would need to measure monoculture densities of individual mutants, then add mutants at the same inoculum into the consortia experiments (now doubling the total bacterial inoculum of the monoculture experiments), and then measure individual strain abundance and/or total biomass to accurately test for facilitation or cooperation (the consortia biomass must exceed the sum of the 2 monoculture biomasses, and then it still might be either exploitation, facilitation, or cooperation, based on the strength of the interactions). I am still very impressed and intrigued by this study and the data obtained, but I do think it needs to focus more on the applied aspects: High performing mutant library consortia to improve agricultural yields. The experiments performed, however, don't tell us much about microbial ecology.

I am quite confused by the clusters. I don't understand the methods of this PCA plot and how mutants are assigned into the various clusters. Perhaps isolates that did not fall within the bounds, or intersected multiple groups should be excluded. The K-means clustering needs significantly more explanation in the methods. As a key aspect of the data being provided, the written methods section should make it very clear how this clustering is being performed and why it is a valid method.

Is there precedent for this type of phenotypic clustering for a transposon library? I think there is tremendous value of having a transposon library of this sort. But I don't really understand the purpose of the clustering.

Line 307-315 – Data are provided for the clusters and compared to the WT. However, statistical tests included in Figures 1E, F measure differences between clusters, rather than each cluster compared to the WT. I see that the WT is provided as the solid black line, but I think adding an additional box plot for the WT would be more appropriate.

Line 321 – I worry that a term like 'specialist mutants' could cause confusion, and perhaps imply that these mutants have niche separation relative to the WT, which hasn't been measured. I think simply 'mutants' would be more appropriate.

Line 332 – It is unclear to me how the predictive performance values were calculated. Please be explicit. When you sum the trait values of monocultures, are you first dividing that trait value according to the reduction in biomass added to the inoculum. ie when you do a 2 species consortium and add each species at 50% of their inoculum as compared to the WT, are you halving the expected trait value? Is there an existing precedent for this?

Line 363 – Stability has a specific definition in ecology that I don't think is intended here. Consider not using this term.

Line 366 – I don't understand the difference between high swarming and swarming motility. These are not differentiated in the methods.

Line 356 – 383 – I'm having a hard time following the data and logic provided here.

Discussion:

Line 387 – Did your results show this? Which data/Figure? I think it would be appropriate to remind the reader where to find the evidence for this claim.

Line 392 – The intra-species diversity also introduces conflict. There is likely more inter-genotype competition in the communities assembled for this study as compared to one involved multi-species communities. The competition/conflict just hasn't been measured here.

The discussion seems overly focused on the individual mutations identified and their resulting phenotypes. Perhaps this level of depth provided for these mutants is more suited for the Results section.

Line 462 – "It is thus possible that consortia were together able to overcome the trait trade-offs experienced at the individual mutant level, leading to improved and more stable ecosystem functioning in time." This language is too vague and has the potential to mislead readers. What is meant by improved ecosystem functioning? Regarding stability, this study does not measure the ecological interactions between different bacterial genotypes. The levels of intraspecific competition between microbes is likely very high, which may lead to increased stability. It may also lead to extinction of the weaker strain.

Line 468 – Is a transposon library really an example of synthetic biology? This is not how I understand the limits of the field, but I apologise if I'm uninformed.

Line 469 – I do not agree that this study highlights the importance of intraspecies diversity in ecosystem functioning. Instead, this study does highlight how intraspecies diversity libraries can be an effective mechanism for identifying traits and consortia which can rapidly improve desired measurables within a defined ecosystem. I think an overall refocusing on the applied potential of the study would be appropriate.*Reviewer #3 (Recommendations for the authors):*

Yang et al. assembled a set of synthetic consortia using functionally specialized mutants generated from a rhizosphere probiotic, Bacillus amyloliquefaciens, and tested whether the consortia can promote the growth of the plant. Because the mutants are derived from the same strain, they should not antagonize their kins. Accordingly, this design potentially reduces within-community antagonism, overcoming a limitation occurring in the previous multi-species consortia.

Although the idea of this paper is of interest to readers in the field of microbiome and plant-microbe interactions, the manuscript is not well-organized. The Introduction section did not adequately discuss the strengths of this design compared to the previous studies. Instead, the author proposed several scientific questions that were not addressed/focused on by this study. In the Results section, several important data are missing, and some claims are not well-supported by the data. The specific comments are listed below.

Introduction:

The author claims that "inoculant design should aim to minimize negative interactions within consortia" (line 46) and the author claimed that three ways (concepts) can be used to achieve this aim: ecological complementarity, division of labor or facilitation between consortia members.

Firstly, the discussion on these concepts (lines 50-75) makes me feel that the author thinks they are three totally segregated concepts (For example, in lines 65-66: "In addition to occupying different niches or performing specialized tasks"). However, there are many overlaps among these three concepts. For example, if different species exploit different niches and such diversification benefits all the individuals, it can be also defined as a "division of labor" (based on definitions in ref. 26). Moreover, many cases of facilitation can be defined as the division of labor (see this ref: https://doi.org/10.1016/j.jmb.2019.06.023). I would suggest that the author use more specific/distinguishable terms throughout the paper, for example, niche complementarity/ specialization within a single niche/ leaky section. In addition, the direct description of how such engineering works (e.g., engineering species to occupy different niches) may be clearer to readers than just giving terms.

Secondly, the authors discussed theories but did not state how the design in this study can benefit from these theories to overcome the proposed limitation. This makes it unclear why constructing such mutant-based synthetic consortia can "minimize negative interactions within consortia". The author proposed a main scientific question "whether this should be based on ecological complementarity, division of labor or facilitation between consortia members. (Lines 74-75)" but this study did not address this question.

Thirdly, following the above point, a logical connection is missing between the last two paragraphs of the Introduction. In other words, does increasing intra-species diversity benefit the ecological complementarity, division of labor, or facilitation, so reduce the within-community antagonism? In addition, I would expect the authors to explain more about why increasing the intra-species diversity of a single bacterium could prevent conflicts between the consortia members. I think this is the main novelty of this study different from the previous design constructing consortia using different species. I saw some reasons were listed in lines 101-103 but expect more explanations/discussions.

In sum, the author could do a better job linking theory and experimental design.

Results

Line 320-322: I think this "antagonism" experiment is very important for the novelty of this study. Because the mutants are derived from the same strain, they should show less competition/antagonism within consortia, which is different from the situation in the previous multi-species consortia. Therefore, (1) the data of the "agar overlay assays" should be provided in the main text of the paper; (2) It will be beneficial to add more evidence (e.g., liquid co-culture); (3) It is also very important to examine if the mutants within a consortium stable coexist during the community assembly. Accordingly, the structure of the consortium during/after the assays (swarming, biofilm formation, root colonization) should be measured.

Design/analyses of the consortia experiment: in lines 324-326, the authors hypothesized that "consortia could show improved performance due to ecological complementarity where different mutants 'specialize' respective to different traits, overcoming trade-offs experienced at individual strain level (Figure 1B)." However, the experiments were concluded with "mutant consortia diversity was positively linked with consortia performance in vivo, which was associated with consortia mean performance and pathogen suppression measured in vitro. (Lines 354-355)" Obviously, the hypothesis was not well examined using the current design and analyses. For a direct test of the hypothesis, the authors should compare the performance of the consortia with niche specialization (e.g., consortium no.29) with that of the others.

Line 383: I think the conclusion here is less evident. To prove the improvement of the consortia functioning is due to the insurance effects, the performance of the 47-member consortium should be compared to the consortium with fewer members. A more rational design is to build consortia according to the four testing functions. Some consortia contain one mutant specializing in one specific function, while the others contain more mutants for the function (that is, add redundancy). Then compare the root colonization and plant protection ability of the two groups.

[Editors’ note: further revisions were suggested prior to acceptance, as described below.]

Thank you for submitting your article "Engineering multifunctional rhizosphere probiotics using consortia of Bacillus amyloliquefaciens transposon insertion mutants" for consideration by *eLife*. Your article has been reviewed by 3 peer reviewers, and the evaluation has been overseen by myself and Detlef Weigel as the Senior Editor.

We had substantial internal discussion of the revised manuscript, and we agree that the revisions have been substantial. All referees would – in principle – support publication of the work in *eLife*.

However, we felt that there is still one important issue that must be carefully considered and addressed. This is regarding the new supernatant experiment, which might not be the best way to infer ecological interactions (please see comments from ref#3) and might lead to misleading interpretations. We agreed that this issue should be clarified before final acceptance of the work. We see two potential options to do so:

1) If you wish to firmly conclude that reduced competition among consortia members exists, please include single strain vs. pairwise co-culture experiments followed by colony counts (see the suggestions below for how to overcome the difficulty that the strains are nearly identical). This might actually require a limited amount of work and would be most appropriate to infer potentially neutral interactions among consortia members.

2) Alternatively, you should remove the supernatant data from the manuscript and adopt a more tempered tone for all assertions linked to how "this study" enhances our understanding of function by mitigating competition between the consortia members.

*Reviewer #1 (Recommendations for the authors):*

The work is significant because it reports that intra-species phenotypic diversity in bacteria could be an effective way to improve probiotic consortia in agriculture. Authors now provide compelling evidence indicating that near-clonal bacteria assembled into consortia based on diverse functional traits perform better than randomly-assembled consortia. Two solid conclusions emerging from this work are that (1) Communities of near-clonal bacteria can be optimized based on functional traits to promote functional diversity and reduce bacterial competition (2) consortium multifunctionality – but not richness – is key to promoting bacterial colonization in roots and to protect against a plant pathogen.

In my opinion, authors have seriously and thoroughly addressed the previous concerns raised by me and other reviewers. I acknowledge the fact that they have performed additional experiments that have strengthened their conclusions. Particularly, they performed supernatant experiments testing pairwise interactions between the mutants included in their phenotypically-optimized 8-member consortium and performed validation experiments with randomly assembled 8-member consortia. These two novel experiments revealed that:

1. Limited negative or positive interactions were observed in supernatant exposure experiments, indicating that bacterial-bacterial interactions are not extensively contributing to the observed phenotypes.

2. Importantly, they used random, 8-member mutant consortia, and demonstrate that the phenotypically-optimized consortium indeed performs better than what is expected by chance. This experiment convinced me that the strategy used in the manuscript has relevance for optimizing synthetic communities based on (multi)-functional traits.

They have also reorganized the structure of the manuscript and rewrote the introduction and Discussion sections to focus more on the methodological novelty of the work and less on ecological theories.*Reviewer #2 (Recommendations for the authors):*

I only have a small suggestion optional to the authors. It might be intriguing to discuss the microscale-level mechanisms behind why the consortium outperforms a single strain. Could the four plant growth-promotion traits act as public goods? For instance, could biofilm formation specialists aid pathogen suppression specialists in better colonizing plants? It may be interesting to add some discussions to the paragraph in lines 497-516.

*Reviewer #3 (Recommendations for the authors):*

Collectively, I am excited to see this manuscript again, as I've been keeping an eye out to see if it had been published in another journal since my first review. It is an intriguing work and I think that many readers in the community will be interested to read it. My primary criticism remains the same as the first review, where ecological interactions are not appropriately measured. I give a detailed explanation of this in Response to R13 (below), along with references that I believe do perform appropriate experiments to measure ecological interactions between strains/genotypes. That said, my other comments appear to be appropriately addressed.

Additionally, because much of the narrative of the discussion has been rewritten to avoid direct claims about ecological interactions between the genotypes throughout the text, I no longer believe that an experiment demonstrating ecological interactions is necessary. Figure S5 should be removed, as this is not an appropriate measure of ecological interactions. The data gathered as part of Figure S5 are likely sufficient to still represent these experiments appropriately. If presented appropriately, I strongly suspect that negative interactions will be the predominant interaction type, so it would be up to the authors if they would like to present this data or not.

So long as reference to ecological interactions (whether neutral or otherwise) and Figure S5 are removed from the text (or ecological interactions are measured according to a method similar to those detailed below), the remainder of this work appears to hold together well, with the text supported by the data.

Response to R13:

Regarding Figure S5. While supernatant assays are not the preferred method to measure ecological interactions, I appreciate the authors performing an experiment to address this major comment from my first review. However, this is not the appropriate experiment and it does not support the claim that interactions between the different genotypes are neutral.

The ideal experiment would be to grow each strain in monoculture and count cfu/mL, then grow each strain in pairwise combination and count cfu/mL of each strain. If it is not possible to selectively plate in order to identify the different genotypes, one could also grow each strain in monoculture and count cfu/mL. Then again repeat growing strains in pairwise co-culture, and count the total cfu/mL. If the interactions are neutral, the prediction is that the total cfu/mL will be the sum of the two strains monoculture cell densities (cfu/mL). One could interpret this as clear niche differentiation, with no interference or exploitative competition. If you are committed to the growth supernatant assay and measuring OD600, you could grow each strain in monoculture, measure final OD600 and then filter the supernatant just as you've done. You can then grow them again in the spend media and measure the final OD600. If the interactions are neutral, then prediction is that growth in monoculture with spent media = growth in monoculture with fresh media. For any strain combination, if growth in monoculture of spent media < growth in monoculture of fresh media, this is a negative interaction.

Unfortunately, it is not satisfactory to infer ecological interactions as a measure of growth in supernatant relative to growth in one's own supernatant. There is full niche overlap when a strain grows in its own supernatant. Using this metric, you could have two different strains with identical nutrient requirements, that will compete with each other for these nutrients, and you would likely measure it as a neutral interaction. It is also challenging when measuring ecological interactions in liquid LB and trying to extrapolate this to interactions in vivo where the host is also present. An experiment like this in liquid LB is probably a good proxy for gauging interactions, but an experiment that is close to the environment of interest would be the most desirable. However, I understand that this can present additional challenges. I also understand that a specific aim of this study is to use in vitro phenotyping in order to optimize consortia functioning. So in that regard, an LB experiment is perhaps appropriate.

I hope that this criticism is clear. For a more comprehensive description of methods in this regard, I suggest Foster and Bell, 2012. DOI: 10.1016/j.cub.2012.08.005. Additionally, Weiss et al. ISME 2022 DOI: 10.1038/s41396-021-01153-z performs nice experiments in this regard, using both monoculture vs coculture experiments (quantifying strain abundances with qPCR) as well as spent supernatant assays.

And yet, after all of this, after reading the revised discussion, I no longer think that this experiment is essential. Because the authors have removed much of the narrative regarding ecological significance and focused more on the applied aspects of this work, I do not think that these ecologically-focussed experiments are completely necessary. So long as the authors avoid claims of the interactions between genotypes (as I would argue these have still not been appropriately measured), then I don't think this experiment is necessary for this manuscript.

---

## [Author Response]

[Editors’ note: the authors resubmitted a revised version of the paper for consideration. What follows is the authors’ response to the first round of review.]

Comments to the Authors:We are sorry to say that, after consultation with the reviewers, we have decided that this work will not be considered further for publication by eLife. Note that while we cannot publish the current study, we remain in principle interested in the work. If you are able to seriously revise the work and respond to the concerns, we would be prepared to review the work again, although we would treat it as a new submission.As you will see, although the reviewers see the work as novel and with the potential to design probiotic consortia, they have also raised a number of issues that will require new experiments, new analyses and important reformatting of the manuscript. After internal discussions with the reviewers, we remain in principle interested in the work but we felt that it is important to give you more time if you wish to thoroughly address the concerns. Particularly, performing pairwise interactions among mutants, testing random sets of 8-member consortia, and reorganizing the manuscript appears to be critical to corroborate the claims and improve clarity, respectively. Furthermore, the reviewers think that the experiments were not designed to test ecological theories regarding ecosystem functioning and stability but rather as a method to design probiotic consortia in agriculture. Many concepts mentioned in the introduction and discussion might not be relevant here and might distract rather than inform.

We thank the Editor and three anonymous reviewers for highly useful feedback on our manuscript. We have now carefully followed all the suggestions and fully revised our manuscript accordingly. The main changes include:

Additional supernatant experiments testing pairwise interactions between the mutants included in ‘phenotypically optimized’, 8-member consortiumAdditional validation experiments with randomly assembled 8-member consortiaReorganization of the structure of the manuscript following reviewers’ suggestionsRewriting of the introduction and discussion and clarification of the aims of the study

We hope that these changes have improved our manuscript and that it could now be reconsidered as a new submission to *eLife*.

Reviewer #1 (Recommendations for the authors):Overall, the concept/methodology is interesting and the results open novel insights towards the engineering of microbial consortia that promote disease resistance. Furthermore, the statistical analyses seem to be conducted and represented adequately. The text is well written and methods are nicely detailed. Despite the novelty, the manuscript has some, yet important weaknesses that will need to be addressed.Strengths:Multi-species inoculants often fail to promote plant health under natural conditions because microbe-microbe interactions are not often considered when consortia are designed and assembled. Therefore, the idea of promoting intra-species multi-functionality through the assembly of bacterial mutants derived from a single strain is exciting. This is because it could promote functional diversity whilst at the same time minimizing the risk of competition between strains. The research question is therefore interesting and novel, the combination of methods, which include strain engineering (Tn5 mutant library), large-scale phenotypic screening and assembly of synthetic microbial consortia is adequate to test the relevance of functional complementary and multi-functionality between mutant strains.The greenhouse experiment using a natural soil is also relevant to test the effects of the inoculants under semi-natural conditions.The statistical analysis is adequate, the manuscript well written and the figures have been nicely crafted.The conclusions are largely supported by experimental evidence, although it remains difficult to unambiguously conclude that the method actually works.

We thank reviewer #1 for positive feedback.

Weaknesses: There are some weaknesses.1. First, the authors used an initial random pool of 479 mutants for in vitro trait phenotyping. It remains unclear why more mutants have not been tested given the fact that these are high-throughput in vitro screens. The authors did not use a saturated mutant library and might have overlooked many important genes.

We have now included more in vitro phenotyping data to the manuscript. The full dataset now includes phenotypic data for 1999 mutants. Of these, we selected 479 phenotypically representative mutants for more detailed analysis and probiotic bioinoculant design. While we likely missed certain mutants with this method, our analysis similarity suggests that this method captured a representative and phenotypically diverse sample of mutants from the original collection. We have now justified this in the manuscript on lines 149-153.

2. Second, the structure of the manuscript is confusing. It remains unclear how the three clusters in Figure 1 have been defined and why they were not considered for the rational selection of the consortia shown in Figure 2. The most confusing part comes from the last paragraph and Figure 3 which shows data based on individual strains and not consortia. This is redundant with the data shown in Figure 1 and this part provides very limited information regarding the "underlying mechanisms between consortia diversity and improved performance". As it is presented this part distracts rather than informs. I do not think that this is the rights way to test the "insurance hypothesis". this hypothesis makes sense if tested in a community context with consortia of different complexities (as in Figure 2).

We have now restructured and clarified these sections. Specifically, we now present all individual strain-based results before the pairwise and consortia data (including new additional data on the effects of randomly assembled 8-mutant consortia). Moreover, we have made the selection of subset of strains from the larger mutant collection clearer (the link between Figures 1 and 2; see also our response to point 1).

3. It remains also unclear whether complementarity between specialized bacterial mutants is really needed for improving plant performance (as stated in the title). The authors showed that the rational selection based on Figure 1 has some limits to predicting community behavior in vitro and in planta. Therefore, a key question is whether microbial consortia composed of random pools of 8 mutants (i.e. not pre-selected based on in vitro phenotypes) would provide similar fitness benefits as the mutant pool selected in Figure 2. In other words, is the first part of the paper really required or could one simply inoculate a pool of Tn5 mutants and observe the same protective effect? Testing the effects of several independent mutant consortia would be needed to draw general conclusions.

This is a very important point. As a result, we have conducted additional experiments where we test the plant protection by randomly assembled 8-mutant consortia that vary in their phenotypic dissimilarity and predicted relative performance. These results confirm that randomly constructed mutant consortia perform worse compared to phenotypically ‘optimized’ consortium, suggesting that improvement was unlikely driven by consortium richness *per se*.

4. Finally, the paper is mechanistically weak. That being said, this is not the core of the manuscript and therefore it is not an issue. Although requesting complementation lines for the 8 mutants used for consortia establishment is beyond the scope of the manuscript, it is important to keep in mind that it is difficult to conclude anything regarding the bacteria genetic determinants identified here without proper functional validations.

We clearly admit these limitations in the discussion and suggest future work to uncover mechanistic aspects of our approach at the genetic level.

5. To strengthen the conclusions shown in Figure 2, it would be important to test other independent consortia of 8 mutants and validate directionality in their effects depending on whether they were selected randomly or based on rational criteria. The concern is that part 1 and part 2 of the manuscripts might not be connected and it would be important to validate that the rational selection based on in vitro phenotypes is actually required and important.

New data has now been included to strengthen these conclusions; please, also see our response to point 3.

6. The results presented in Figure 3 (individual strains) are insufficient to test the insurance effect. One would need to inspect this in the context of the various consortia shown in Figure 2. Here, it would be important to assess 1. total Bacillus load (e.g. using colony counts), 2. disease suppression and 3. "community structure" of the bacilli mutants. The latter can likely easily be conducted using PCRs with pairs of Tn5 transposon specific and random primers similar to classical TnSeq approaches. If the insurance hypothesis holds true, one would expect, that throughout the whole growth cycle, the complex community always performs better than the individual mutants (based on colony counts and disease suppressiveness) and also that relative abundances of Bacilli mutants change across the growth cycle. Such experiments could further reinforce the conclusion regarding insurance effects. Alternatively, authors should consider including this section as Figure 2 and tone-down their claim regarding these insurance effects that are not corroborated by experimental evidence.

We agree with the reviewer that more detailed measurements and quantification of each mutant frequency would be required to support our conclusions on the insurance hypothesis. As the Tn insertions did not include unique barcodes, quantifying mutant frequencies was not possible (now mentioned in the discussion). We have hence toned down our conclusions on the role of insurance hypothesis and have removed it from the introduction as our hypothesis.

7. Agar overlay data not shown?

Agar overlay data is now included. We have also added new data on the lack of pairwise inhibitory effects between the strains based on supernatant growth assays (see revised methods on lines 375-378).

Reviewer #2 (Recommendations for the authors):This study provides a novel, potentially high-throughput method for identifying beneficial microbial consortia towards plant growth and health, which may be applicable towards other biotechnological applications. This study has the potential to be of great interest to all those working in host-microbe interactions. However, it is my opinion that there is an overemphasis on microbial ecology and the inherent value of intraspecies diversity towards ecosystem health, where I think the data of this study does not meet their intended aims.Strengths: This study provides a novel and unique approach for studying the impact of intraspecies diversity on microbial consortia and understanding host (plant)-microbe interactions. There is a very good introductory section highlighting the field of microbial ecology. The methods used seem applicable to many different potential environments, as highlighted by the authors, and might be of great interest to those working on both applied and fundamental areas of microbiology and/or host-microbe interactions. The justification of the study, the introduction, and the methods used are all very high quality. The stated expectations by the authors are also well-supported by the theoretical literature and clearly explained. The focus on intraspecies interactions and diversity is also very welcomed and appears to me to be a novel approach.

We thank reviewer #2 for positive comments.

Weaknesses: It is my opinion that the narrative in the results and discussion is overly focused on the impact of intraspecies diversity in microbial ecology, and how this diversity improves ecosystem functioning. I do not think the study provides support for these claims. Instead, the study does provide excellent support for high-performing consortia of mutants, which outperform the WT monoculture across varying metrics, which has both applied and fundamental value regarding agriculture and host-microbe interactions at both the strain and molecular levels. The data provided as apart of this work provides many new avenues of research and I think is an excellent step towards developing beneficial probiotic consortia. Unfortunately, these advances are sometimes overshadowed by claims of ecosystem stability and ecosystem functioning that lack support in the data. In multiple places in the text, it is suggested that intraspecies consortia are not subject to intra-species conflict. I find this to be potentially misleading, as it implies that these genotypes now have neutral (or positive) interactions, which is not appropriately measured nor expected from the data.

We have now toned down our conclusions on the effects of intra-species diversity for the ecosystem functioning and stability. To provide more support for the lack of inhibitory effects between mutants, we have included new data based on supernatant growth assays, which demonstrates that most of the pairwise interactions were neutral or mildly positive or negative (see updated Figure S5).

Line 44-47 – I am unsure of the support that antagonistic traits often increase invasion success of probiotic bacteria, or what this statement really means. I think it could benefit by being more explicit.

We have now replaced ‘invasiveness’ with ‘competitiveness’ and have expanded the example to be more specific.

The introduction may benefit from a more thorough explanation of biodiversity-ecosystem functioning theory (line 84), to match the excellent introduction of microbial ecology from lines 50-75.

We have now clarified the biodiversity-ecosystem functioning approaches earlier during the introduction to make the key idea clearer (on lines 55-83).

Line 95: 'benefits' to 'beneficial

Revised as suggested.

Line 96: Expected advances. It does not appear that expectations 2 or 3 are tested. Expectation 3 is also rather unclear. Please consider rewriting this section to better set the reader's expectations for the data and results in the paper.

This section has now been simplified and completely rewritten.

Line 162 – Please clarify how it is known that OD600 = 0.05 equals ~10^5 cells/mL for B. amyloliquifaciens. Or omit the ~10^5 cells/mL. Additionally, I wonder if OD600 is an appropriate proxy for biomass production in a B. amyloliquifaciens mutant library? Plating and counting CFU/mL would be the much more reliable measurement. Considering the method used for biomass production measurement, did you perform these measurements in a plate reader and have additional time series data beyond the 24hr timepoint? Perhaps maximum growth rate or other factors of the growth curve may also be good indicators of beneficial phenotypes for plant health.

The relationship between the OD and cell numbers is based on a calibration curves, which we now present for the wild-type, and low and high biomass producing mutants in Figure S1.

General Methods Comments: Given the more applied aims of this study, maintaining a constant inoculum size and measuring the outputs of the consortium is a very reasonable strategy. However, performing experiments in this way ignores the intraspecies competition that likely defines the system. This makes claims of facilitation, exploitation, or cooperation moot, as they are not being appropriately measured. The authors would need to measure monoculture densities of individual mutants, then add mutants at the same inoculum into the consortia experiments (now doubling the total bacterial inoculum of the monoculture experiments), and then measure individual strain abundance and/or total biomass to accurately test for facilitation or cooperation (the consortia biomass must exceed the sum of the 2 monoculture biomasses, and then it still might be either exploitation, facilitation, or cooperation, based on the strength of the interactions). I am still very impressed and intrigued by this study and the data obtained, but I do think it needs to focus more on the applied aspects: High performing mutant library consortia to improve agricultural yields. The experiments performed, however, don't tell us much about microbial ecology.

As we were interested in separating richness and species identity effects in mutant consortia, we employed substitutive experimental design, which assumes that the biomass of the consortia is kept constant (increasing individual strain density in consortia would make it impossible to disentangle diversity versus abundance effects). We now acknowledge in the discussion that our method also affected the mutant abundances relative to monoculture controls, which could have affected the observed patterns. We have now included additional data for analyzing interactions between a subset of mutants used in the diversity-ecosystem functioning experiment (see response to point 7). The applied aspect of the study has now been emphasized.

I am quite confused by the clusters. I don't understand the methods of this PCA plot and how mutants are assigned into the various clusters. Perhaps isolates that did not fall within the bounds, or intersected multiple groups should be excluded. The K-means clustering needs significantly more explanation in the methods. As a key aspect of the data being provided, the written methods section should make it very clear how this clustering is being performed and why it is a valid method.Is there precedent for this type of phenotypic clustering for a transposon library? I think there is tremendous value of having a transposon library of this sort. But I don't really understand the purpose of the clustering.

We have now clarified the K-means clustering in the methods (lines 186-189). Briefly, K-means clustering assigns *n* observations into *k* clusters where each observation (in our case mutant) belongs to a cluster with the nearest mean (cluster centers or cluster centroid). PCA analysis was then used to visualize the phenotypic clustering where the bounds of the clusters indicate 95% confidence intervals. Each mutant was hence assigned to one of the clusters even though mutants varied in their distance to cluster centroid means. This method is widely used in ecological studies with microbes and other taxa.

Line 307-315 – Data are provided for the clusters and compared to the WT. However, statistical tests included in Figures 1E, F measure differences between clusters, rather than each cluster compared to the WT. I see that the WT is provided as the solid black line, but I think adding an additional box plot for the WT would be more appropriate.

We have now clarified in panels Figure 1E-F and results text that WT was assigned in the cluster 1. As the WT is only one clone, variation between technical replicates is very small. We hence feel that presenting WT performance as solid black line offers better visualization of the WT baseline. We have now included shaded area around WT mean to show variation between technical replicates.

Line 321 – I worry that a term like 'specialist mutants' could cause confusion, and perhaps imply that these mutants have niche separation relative to the WT, which hasn't been measured. I think simply 'mutants' would be more appropriate.

We have now simplified the text as suggested and only use the term ‘Specialist mutant’ when it is related to ecological specialism or generalism. When emphasizing phenotypic differences between mutants, we have now phrased them as ‘phenotypically distinct’ or ‘phenotypically dissimilar’ mutants depending on the context.

Line 332 – It is unclear to me how the predictive performance values were calculated. Please be explicit. When you sum the trait values of monocultures, are you first dividing that trait value according to the reduction in biomass added to the inoculum. ie when you do a 2 species consortium and add each species at 50% of their inoculum as compared to the WT, are you halving the expected trait value? Is there an existing precedent for this?

This has now been clarified in the methods; predicted value equals to the sum of trait values of each member divided by the richness of given consortium based on additive model. As a result, we are not halving the trait values as the chosen growth-based traits. These potential inoculum abundance effects are now briefly mentioned in the discussion.

Line 363 – Stability has a specific definition in ecology that I don't think is intended here. Consider not using this term.

We have removed the references to stability here.

Line 366 – I don't understand the difference between high swarming and swarming motility. These are not differentiated in the methods.

Revised to remove ambiguity – with “high” we just referred to relatively high swarming motility trait values.

Line 356 – 383 – I'm having a hard time following the data and logic provided here.

This section has now been rewritten to improve clarity.

Discussion:Line 387 – Did your results show this? Which data/Figure? I think it would be appropriate to remind the reader where to find the evidence for this claim.

Reference to specific results now included as suggested.

Line 392 – The intra-species diversity also introduces conflict. There is likely more inter-genotype competition in the communities assembled for this study as compared to one involved multi-species communities. The competition/conflict just hasn't been measured here.The discussion seems overly focused on the individual mutations identified and their resulting phenotypes. Perhaps this level of depth provided for these mutants is more suited for the Results section.

We now discuss potential conflicts of intra-species diversity in the broader context of bioinoculant design beyond this experiment. We feel that discussing individual mutations is still appropriate to provide some potential mechanistic explanations for our results and have kept them in the discussion.

Line 462 – "It is thus possible that consortia were together able to overcome the trait trade-offs experienced at the individual mutant level, leading to improved and more stable ecosystem functioning in time." This language is too vague and has the potential to mislead readers. What is meant by improved ecosystem functioning? Regarding stability, this study does not measure the ecological interactions between different bacterial genotypes. The levels of intraspecific competition between microbes is likely very high, which may lead to increased stability. It may also lead to extinction of the weaker strain.

Revised to add clarity; with “ecosystem functioning” we refer to “plant protection” in this occasion. With stability we here refer to stability of plant protection – from the microbial point of view this could be mediated by lack or intense competition, which we have now clarified. We have now revised this section to reduce ambiguity of the terms used.

Line 468 – Is a transposon library really an example of synthetic biology? This is not how I understand the limits of the field, but I apologise if I'm uninformed.

Reference to ‘synthetic biology’ removed.

Line 469 – I do not agree that this study highlights the importance of intraspecies diversity in ecosystem functioning. Instead, this study does highlight how intraspecies diversity libraries can be an effective mechanism for identifying traits and consortia which can rapidly improve desired measurables within a defined ecosystem. I think an overall refocusing on the applied potential of the study would be appropriate.

Revised as suggested; we now emphasize our findings in more applied context as suggested and have toned down the significance of intra-species diversity.

Reviewer #3 (Recommendations for the authors):Yang et al. assembled a set of synthetic consortia using functionally specialized mutants generated from a rhizosphere probiotic, Bacillus amyloliquefaciens, and tested whether the consortia can promote the growth of the plant. Because the mutants are derived from the same strain, they should not antagonize their kins. Accordingly, this design potentially reduces within-community antagonism, overcoming a limitation occurring in the previous multi-species consortia.Although the idea of this paper is of interest to readers in the field of microbiome and plant-microbe interactions, the manuscript is not well-organized. The Introduction section did not adequately discuss the strengths of this design compared to the previous studies. Instead, the author proposed several scientific questions that were not addressed/focused on by this study. In the Results section, several important data are missing, and some claims are not well-supported by the data. The specific comments are listed below.

We thank Reviewer #3 for highly useful comments.

Introduction:The author claims that "inoculant design should aim to minimize negative interactions within consortia" (line 46) and the author claimed that three ways (concepts) can be used to achieve this aim: ecological complementarity, division of labor or facilitation between consortia members.Firstly, the discussion on these concepts (lines 50-75) makes me feel that the author thinks they are three totally segregated concepts (For example, in lines 65-66: "In addition to occupying different niches or performing specialized tasks"). However, there are many overlaps among these three concepts. For example, if different species exploit different niches and such diversification benefits all the individuals, it can be also defined as a "division of labor" (based on definitions in ref. 26). Moreover, many cases of facilitation can be defined as the division of labor (see this ref: https://doi.org/10.1016/j.jmb.2019.06.023). I would suggest that the author use more specific/distinguishable terms throughout the paper, for example, niche complementarity/ specialization within a single niche/ leaky section. In addition, the direct description of how such engineering works (e.g., engineering species to occupy different niches) may be clearer to readers than just giving terms.

These are very good points, and we have now made it clearer in the introduction that these processes are not mutually exclusive. We have also clarified the terminology and rewritten the whole introduction and reduced the emphasis on division of labor and complementarity, which we did not directly quantify in our experiments.

Secondly, the authors discussed theories but did not state how the design in this study can benefit from these theories to overcome the proposed limitation. This makes it unclear why constructing such mutant-based synthetic consortia can "minimize negative interactions within consortia". The author proposed a main scientific question "whether this should be based on ecological complementarity, division of labor or facilitation between consortia members. (Lines 74-75)" but this study did not address this question.

We have now made our research questions clearer at the end of the introduction.

Thirdly, following the above point, a logical connection is missing between the last two paragraphs of the Introduction. In other words, does increasing intra-species diversity benefit the ecological complementarity, division of labor, or facilitation, so reduce the within-community antagonism? In addition, I would expect the authors to explain more about why increasing the intra-species diversity of a single bacterium could prevent conflicts between the consortia members. I think this is the main novelty of this study different from the previous design constructing consortia using different species. I saw some reasons were listed in lines 101-103 but expect more explanations/discussions.

We have now rewritten this section of the introduction and give a better justification why we would expect more closely related communities to be less competitive based on kin selection theory (with appropriate references).

In sum, the author could do a better job linking theory and experimental design.

This is a fair point, and we hope that our revised version makes this link more obvious.

ResultsLine 320-322: I think this "antagonism" experiment is very important for the novelty of this study. Because the mutants are derived from the same strain, they should show less competition/antagonism within consortia, which is different from the situation in the previous multi-species consortia. Therefore, (1) the data of the "agar overlay assays" should be provided in the main text of the paper; (2) It will be beneficial to add more evidence (e.g., liquid co-culture); (3) It is also very important to examine if the mutants within a consortium stable coexist during the community assembly. Accordingly, the structure of the consortium during/after the assays (swarming, biofilm formation, root colonization) should be measured.

More data has now been included (agar overlay and liquid supernatant assays; see response to point 7). Unfortunately, we were not able to quantify changes in mutant frequencies during the experiments as transposon insertions did not include barcodes. We have now made this limitation clear in the discussion.

Design/analyses of the consortia experiment: in lines 324-326, the authors hypothesized that "consortia could show improved performance due to ecological complementarity where different mutants 'specialize' respective to different traits, overcoming trade-offs experienced at individual strain level (Figure 1B)." However, the experiments were concluded with "mutant consortia diversity was positively linked with consortia performance in vivo, which was associated with consortia mean performance and pathogen suppression measured in vitro. (Lines 354-355)" Obviously, the hypothesis was not well examined using the current design and analyses. For a direct test of the hypothesis, the authors should compare the performance of the consortia with niche specialization (e.g., consortium no.29) with that of the others.

Please, see our response to point 3. Briefly, we have now conducted additional experiments where we test the plant protection by randomly assembled 8-mutant consortia. These results confirm that randomly constructed mutant consortia perform worse compared to phenotypically ‘optimized’ consortium.

Line 383: I think the conclusion here is less evident. To prove the improvement of the consortia functioning is due to the insurance effects, the performance of the 47-member consortium should be compared to the consortium with fewer members. A more rational design is to build consortia according to the four testing functions. Some consortia contain one mutant specializing in one specific function, while the others contain more mutants for the function (that is, add redundancy). Then compare the root colonization and plant protection ability of the two groups.

Please, see our response to point 3 and above.

[Editors’ note: further revisions were suggested prior to acceptance, as described below.]

The reviewers have discussed their reviews with one another, and the Reviewing Editor has drafted this to help you prepare a revised submission.We had substantial internal discussion of the revised manuscript, and we agree that the revisions have been substantial.However, we felt that there is still one important issue that must be carefully considered and addressed. This is regarding the new supernatant experiment, which might not be the best way to infer ecological interactions (please see comments from ref#3) and might lead to misleading interpretations. We agreed that this issue should be clarified before final acceptance of the work. We see two potential options to do so:1) If you wish to firmly conclude that reduced competition among consortia members exists, please include single strain vs. pairwise co-culture experiments followed by colony counts (see the suggestions below for how to overcome the difficulty that the strains are nearly identical). This might actually require a limited amount of work and would be most appropriate to infer potentially neutral interactions among consortia members.2) Alternatively, you should remove the supernatant data from the manuscript and adopt a more tempered tone for all assertions linked to how "this study" enhances our understanding of function by mitigating competition between the consortia members.

We thank the Editor and three anonymous reviewers for highly useful feedback on our manuscript. We have now carefully followed all the suggestions and fully revised our manuscript accordingly. While we have not been able to collect more data using CFU counts, we have re-analyzed the data using 50% LB media as control baseline as suggested by the reviewer #3.

The main changes include:

Discussion behind the reasons why consortia outperform single strains.Reanalysis of pairwise mutant interactions using 50% LB media as the baseline control.Toning down our interpretation on the competition between consortia members throughout the manuscript; supernatant data is used to test if mutants inhibit each other’s growth and references to competition have been removed.

Reviewer #2 (Recommendations for the authors):I only have a small suggestion optional to the authors. It might be intriguing to discuss the microscale-level mechanisms behind why the consortium outperforms a single strain. Could the four plant growth-promotion traits act as public goods? For instance, could biofilm formation specialists aid pathogen suppression specialists in better colonizing plants? It may be interesting to add some discussions to the paragraph in lines 497-516.

We thank reviewer #2 for positive and very helpful comments. The production of public goods could indeed play a role here and we now mention this on lines 524-529.

Reviewer #3 (Recommendations for the authors):Collectively, I am excited to see this manuscript again, as I've been keeping an eye out to see if it had been published in another journal since my first review. It is an intriguing work and I think that many readers in the community will be interested to read it. My primary criticism remains the same as the first review, where ecological interactions are not appropriately measured. I give a detailed explanation of this in Response to R13 (below), along with references that I believe do perform appropriate experiments to measure ecological interactions between strains/genotypes. That said, my other comments appear to be appropriately addressed.Additionally, because much of the narrative of the discussion has been rewritten to avoid direct claims about ecological interactions between the genotypes throughout the text, I no longer believe that an experiment demonstrating ecological interactions is necessary. Figure S5 should be removed, as this is not an appropriate measure of ecological interactions. The data gathered as part of Figure S5 are likely sufficient to still represent these experiments appropriately. If presented appropriately, I strongly suspect that negative interactions will be the predominant interaction type, so it would be up to the authors if they would like to present this data or not.

We thank reviewer #3 for positive and very helpful comments. We have now replaced the Figure 3 —figure supplement 2 with a revised one, where we use 50% LB as the baseline control to calculate the relative growth as following formula:

Relative growth=(OD600 sup−OD600 LB)OD600 LB×100%;

OD_600 sup_ and OD_600 LB_ denote for mutants’ growth in other mutants’ supernatant or in 50% LB after 24h of growth (on lines 691-694).

The results show that the biomass production of most strains decreased in the supernatant compared to their growth in 50% LB. While most of these effects were negative (up to 12.7% in magnitude), also some positive effects were observed (up to 7.1% in magnitude). This data has been kept in the manuscript to simply test if insertions affected how mutants shape each other’s growth and references to competition have been removed as suggested.

So long as reference to ecological interactions (whether neutral or otherwise) and Figure S5 are removed from the text (or ecological interactions are measured according to a method similar to those detailed below), the remainder of this work appears to hold together well, with the text supported by the data.Response to R13:Regarding Figure S5. While supernatant assays are not the preferred method to measure ecological interactions, I appreciate the authors performing an experiment to address this major comment from my first review. However, this is not the appropriate experiment and it does not support the claim that interactions between the different genotypes are neutral.The ideal experiment would be to grow each strain in monoculture and count cfu/mL, then grow each strain in pairwise combination and count cfu/mL of each strain. If it is not possible to selectively plate in order to identify the different genotypes, one could also grow each strain in monoculture and count cfu/mL. Then again repeat growing strains in pairwise co-culture, and count the total cfu/mL. If the interactions are neutral, the prediction is that the total cfu/mL will be the sum of the two strains monoculture cell densities (cfu/mL). One could interpret this as clear niche differentiation, with no interference or exploitative competition. If you are committed to the growth supernatant assay and measuring OD600, you could grow each strain in monoculture, measure final OD600 and then filter the supernatant just as you've done. You can then grow them again in the spend media and measure the final OD600. If the interactions are neutral, then prediction is that growth in monoculture with spent media = growth in monoculture with fresh media. For any strain combination, if growth in monoculture of spent media < growth in monoculture of fresh media, this is a negative interaction.

We have now used 50% LB as the baseline control to determine how mutants’ supernatants affect each other’s growth in Figure 3 —figure supplement 2.

Unfortunately, it is not satisfactory to infer ecological interactions as a measure of growth in supernatant relative to growth in one's own supernatant. There is full niche overlap when a strain grows in its own supernatant. Using this metric, you could have two different strains with identical nutrient requirements, that will compete with each other for these nutrients, and you would likely measure it as a neutral interaction. It is also challenging when measuring ecological interactions in liquid LB and trying to extrapolate this to interactions in vivo where the host is also present. An experiment like this in liquid LB is probably a good proxy for gauging interactions, but an experiment that is close to the environment of interest would be the most desirable. However, I understand that this can present additional challenges. I also understand that a specific aim of this study is to use in vitro phenotyping in order to optimize consortia functioning. So in that regard, an LB experiment is perhaps appropriate.

We fully agree with the reviewer that in vivo environment would be the most appropriate way to test the effect of competition between mutants. Hence, we have toned down our interpretation on the competition between consortia members throughout the manuscript.

I hope that this criticism is clear. For a more comprehensive description of methods in this regard, I suggest Foster and Bell, 2012. DOI: 10.1016/j.cub.2012.08.005. Additionally, Weiss et al. ISME 2022 DOI: 10.1038/s41396-021-01153-z performs nice experiments in this regard, using both monoculture vs coculture experiments (quantifying strain abundances with qPCR) as well as spent supernatant assays.And yet, after all of this, after reading the revised discussion, I no longer think that this experiment is essential. Because the authors have removed much of the narrative regarding ecological significance and focused more on the applied aspects of this work, I do not think that these ecologically-focussed experiments are completely necessary. So long as the authors avoid claims of the interactions between genotypes (as I would argue these have still not been appropriately measured), then I don't think this experiment is necessary for this manuscript.

We hope that our revisions have now resolved this final outstanding issue. We also want to thank the reviewer for raising this important point and taking time to thoroughly revise our manuscript.